# A warm Neptune's methane reveals core mass and vigorous atmospheric mixing

David K. Sing[1,2]✉, Zafar Rustamkulov[1], Daniel P. Thorngren[2], Joanna K. Barstow[3], Pascal Tremblin[4,5], Catarina Alves de Oliveira[6], Tracy L. Beck[7], Stephan M. Birkmann[6], Ryan C. Challener[8], Nicolas Crouzet[9], Néstor Espinoza[7], Pierre Ferruit[6], Giovanna Giardino[10], Amélie Gressier[7], Elspeth K. H. Lee[11], Nikole K. Lewis[8], Roberto Maiolino[12], Elena Manjavacas[2,13], Bernard J. Rauscher[14], Marco Sirianni[15] & Jeff A. Valenti[7]

Observations of transiting gas giant exoplanets have revealed a pervasive depletion of methane[1-4], which has only recently been identified atmospherically[5,6]. The depletion is thought to be maintained by disequilibrium processes such as photochemistry or mixing from a hotter interior[7-9]. However, the interiors are largely unconstrained along with the vertical mixing strength and only upper limits on the $CH_4$ depletion have been available. The warm Neptune WASP-107b stands out among exoplanets with an unusually low density, reported low core mass[10], and temperatures amenable to $CH_4$, though previous observations have yet to find the molecule[2,4]. Here we present a JWST-NIRSpec transmission spectrum of WASP-107b that shows features from both $SO_2$ and $CH_4$ along with $H_2O$, $CO_2$, and CO. We detect methane with $4.2\sigma$ significance at an abundance of $1.0 \pm 0.5$ ppm, which is depleted by 3 orders of magnitude relative to equilibrium expectations. Our results are highly constraining for the atmosphere and interior, which indicate the envelope has a super-solar metallicity of $43 \pm 8 \times$ solar, a hot interior with an intrinsic temperature of $T_{int} = 460 \pm 40$ K, and vigorous vertical mixing which depletes $CH_4$ with a diffusion coefficient of $K_{zz} = 10^{11.6\pm0.1}$ cm$^2$ s$^{-1}$. Photochemistry has a negligible effect on the $CH_4$ abundance but is needed to account for the $SO_2$. We infer a core mass of $11.5^{+3.0}_{-3.6} M_\oplus$, which is much higher than previous upper limits[10], releasing a tension with core-accretion models[11].

We observed one transit of the exoplanet WASP-107b (ref. 12) with the G395H spectral grating of the James Webb Telescope Near-Infrared Spectrograph (JWST-NIRSpec) as part of Guaranteed Time Observations (GTO) programme 1224 (principal investigator S. Birkmann). In the first years of operation of the James Webb Space Telescope (JWST), this mode demonstrates reliable detections of $H_2O$, CO, $CO_2$ and $SO_2$ for several giant exoplanets[7,13-15]. See the Methods for further details on the observations and data analysis. The wavelength-integrated JWST-NIRSpec time-series photometry of WASP-107b is shown in Fig. 1, and the transmission spectrum is shown in Fig. 2. The spectrum shows several molecular absorption signals that are readily identifiable. We detect a large $CO_2$ absorption feature at 4.3 μm (ref. 15) (Fig. 2) and a slope-like feature at the shortest wavelengths from $H_2O$ (ref. 14).

We ran a series of model retrievals to provide detailed constraints on the atmospheric composition and temperature (Methods). The $H_2O$ and $CO_2$ features are detected at high confidence (Extended Data Table 2). The spectrum shows CO features between 4.5 μm and 5 μm ($4\sigma$), and an $SO_2$ absorption feature at 4.0 μm ($5.5\sigma$). $SO_2$ is a known by-product of photochemistry[7], which is generated when stellar ultra-violet radiation reacts with $H_2S$ and has been identified in WASP-107b at longer wavelengths with JWST/MIRI[4]. Finally, a narrow $CH_4$ feature is identified at 3.32 μm ($4.2\sigma$). Although $CH_4$ has been unobservable at both shorter and longer wavelengths[2,4], the feature here is the Q-branch band head of $CH_4$, which is strong enough to appear above the clouds and $H_2O$ (Fig. 2).

For $H_2O$, we find volume-mixing ratio abundances of $10^{-1.85\pm0.22}$, which indicates super-solar abundances near 40× solar. The presence of $CO_2$ is known to be a tracer of high metallicity[15,16], and the retrieved abundances of $10^{-3.33\pm0.27}$ support this, although these values are lower than the chemical equilibrium expectations by about an order of magnitude as the molecule is further sensitive to atmospheric mixing. $CH_4$ is also found to be heavily depleted, with retrieved abundances of $10^{-6.03\pm0.21}$. These $CH_4$ abundances are consistent with previous Hubble Space Telescope (HST) and JWST upper limits[2,4]. At the millibar pressures probed in transmission spectra, at 40× solar the atmosphere in equilibrium would be about one part in a thousand

[1]Department of Earth and Planetary Sciences, Johns Hopkins University, Baltimore, MD, USA. [2]Department of Physics and Astronomy, Johns Hopkins University, Baltimore, MD, USA. [3]School of Physical Sciences, The Open University, Milton Keynes, UK. [4]Université Paris-Saclay, UVSQ, CEA, Maison de la Simulation, Gif-sur-Yvette, France. [5]Université Paris-Saclay, Université Paris Cité, CEA, CNRS, AIM, Gif-sur-Yvette, France. [6]European Space Agency, European Space Astronomy Centre, Madrid, Spain. [7]Space Telescope Science Institute, Baltimore, MD, USA. [8]Department of Astronomy and Carl Sagan Institute, Cornell University, Ithaca, NY, USA. [9]Leiden Observatory, Leiden University, Leiden, The Netherlands. [10]ATG Europe for the European Space Agency, ESTEC, Noordwijk, The Netherlands. [11]Center for Space and Habitability, University of Bern, Bern, Switzerland. [12]The Old Schools, University of Cambridge, Cambridge, UK. [13]AURA for the European Space Agency (ESA), Space Telescope Science Institute, Baltimore, MD, USA. [14]NASA Goddard Space Flight Center, Greenbelt, MD, USA. [15]European Space Agency (ESA) Office, Space Telescope Science Institute, Baltimore, MD, USA. ✉e-mail: dsing@jhu.edu

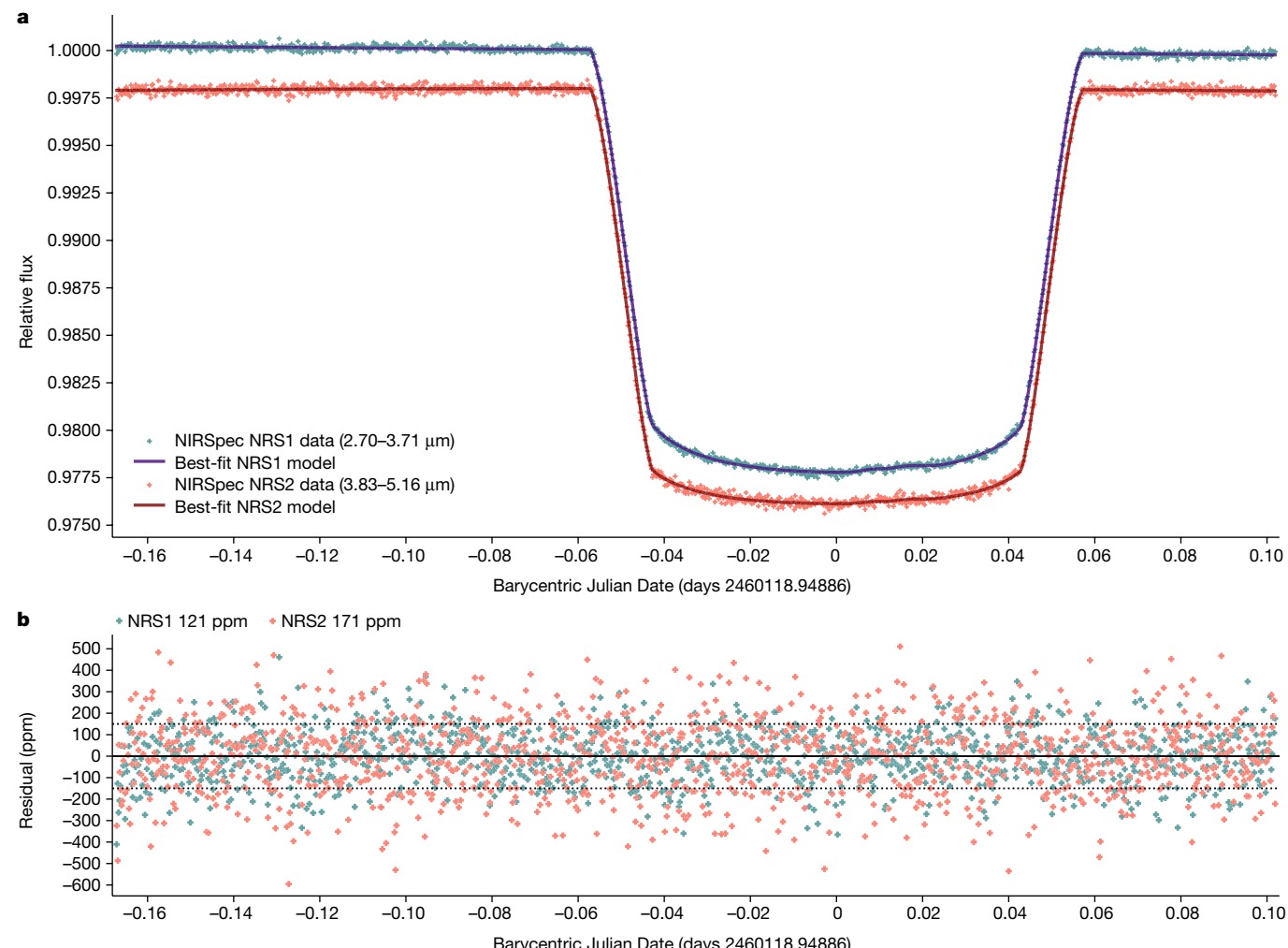

**Fig. 1 | The light curve of WASP-107b observed by JWST-NIRSpec G395H.**
**a**, The normalized wavelength-integrated white-light curves for the two detectors are shown, with the NRS1 (2.70–3.71 μm) and NRS2 (3.83–5.16 μm) detectors offset for clarity. A best-fit limb-darkened transit model is overplotted (blue). See Extended Data Fig. 2 for further details. **b**, The model residuals that achieve near-photon limited precisions.

$CH_4$, whereas abundances slightly less than 1 part per million (ppm) are found. Measuring the amount of $CH_4$ depletion is highly constraining for non-equilibrium models[8], as it can be tied to the pressure and temperature of the hotter interior from which the species is mixed to higher levels. Although not every planet shows a $CH_4$ depletion[5], for those that do such as WASP-107b (ref. 4) and others[3], only the upper limits on the depletion are available. By quantifying the $CH_4$ depletion, the non-equilibrium chemistry can be tightly constrained and the depletion mechanisms can be explored. We find abundances of $SO_2$ to be $10^{-5.06\pm0.13}$, which broadly match the JWST/MIRI results[4] as well as those of WASP-39 b (refs. 7,17) indicating ongoing photochemistry in the atmosphere.

Given the retrieved molecular abundances, we ran a grid of forward atmospheric models to place constraints on the photochemistry, vertical mixing, metallicity and temperature structure of the planet. The model includes non-equilibrium chemistry from both vertical mixing and photochemistry self-consistently calculated[18,19] (Methods). The chemistry was computed using $T$–$P$ profiles in radiative–convective equilibrium, with a range of intrinsic temperatures, $T_{int}$. The intrinsic temperature is related to the emitted flux generated from the interior of the planet, which passes through the atmosphere, with Jupiter having a $T_{int}$ near 100 K. With vertical mixing, most of the molecular species detected are expected to be uniformly mixed from their quench pressures at deeper and hotter conditions in chemical equilibrium to higher altitudes. We modelled the vertical mixing in a turbulent flow using the vertical eddy diffusion coefficient, $K_{zz}$. Our best-fit forward model to the retrieved abundances is shown in Fig. 3 (see also Extended Data Fig. 7). We find the combination of $H_2O$, CO, $CO_2$, $CH_4$ and $SO_2$ to be highly constraining. $H_2O$ and CO are insensitive to non-equilibrium chemistry in this temperature regime (Fig. 3) helping to uniquely measure the metallicity, whereas the combination of $CO_2$ and $CH_4$ are jointly sensitive to $K_{zz}$ and $T_{int}$. $SO_2$ also helps to provide an upper bound on $K_{zz}$ (Extended Data Fig. 7). With these combined constraints, the forward model grid indicates the planet has a hot intrinsic temperature of $T_{int} = 460 \pm 40$ K, a high metallicity of $Z = 43 \pm 8\times$ solar and vigorous vertical mixing with a $K_{zz} = 10^{11.6\pm0.1}$ cm$^2$ s$^{-1}$.

Theoretical models have estimated the strength of vertical mixing and $K_{zz}$ (refs. 20–22). However, the overall uncertainty on $K_{zz}$ for giant planets remains large[23], with values often ranging from $10^4$ cm$^2$ s$^{-1}$ to $10^{12}$ cm$^2$ s$^{-1}$ for hot Jupiters[8]. The empirical constraints on $K_{zz}$ for exoplanets currently rely on broadband Spitzer/IRAC photometry[9], in which there is considerable planet-to-planet stochasticity and the molecular features are spectroscopically unresolved, giving rise to modelling degeneracies. The $K_{zz}$ measurement in WASP-107b here is a substantial improvement over the order of magnitude estimates previously explored using either the upper limits on $CH_4$ from JWST/MIRI[4] or population-level results[9]. The measurement is one of the

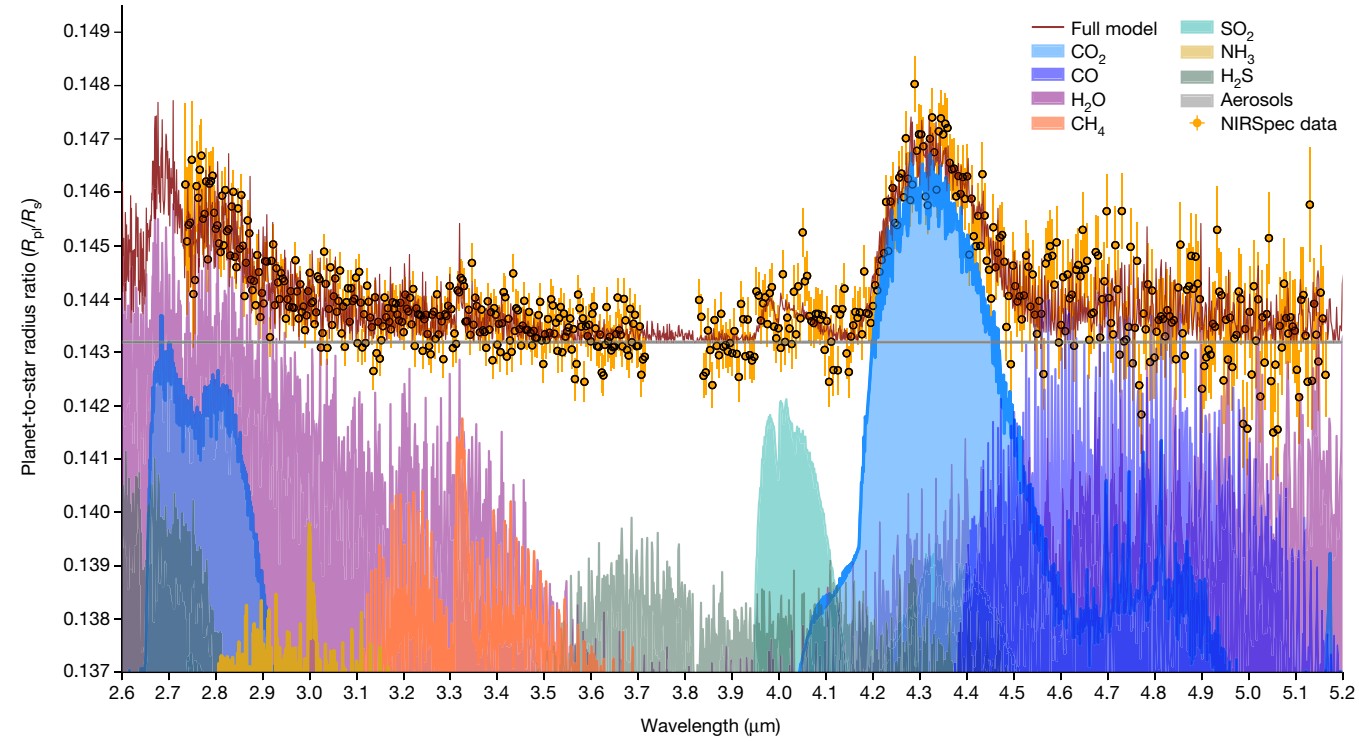

**Fig. 2 | WASP-107b transmission spectral measurements.** JWST-NIRSpec transmission spectrum and the 1$\sigma$ uncertainties. The best-fit ATMO[32] model is also plotted, and the individual contributions for each molecular species are shown.

best evidence so far that non-equilibrium chemistry from vertical mixing is an important physical process in exoplanetary atmospheres, a mechanism long known to be important for Jupiter[24]. The $K_{zz}$ value for WASP-107b at 1 bar is about 1,000–10,000 times higher than the typical estimates used[7,25], which are largely based on three-dimensional general circulation models (GCM)[20]. This could indicate the planet has anomalously high vertical mixing, perhaps because of the high $T_{int}$ resulting in a convective region that reaches

lower than expected pressures. Alternatively, the GCMs may be inadequately capturing the eddy dissipation with the artificial viscosity parameters used[26].

The high $T_{int}$ is a direct constraint on the energy stored in the deep atmosphere, which can help us understand the mechanism responsible for the inflated radius of the planet. The anomalously large radii of irradiated warm Neptunes and hot Jupiters are one of the most intriguing and long-standing problems in our understanding of extrasolar

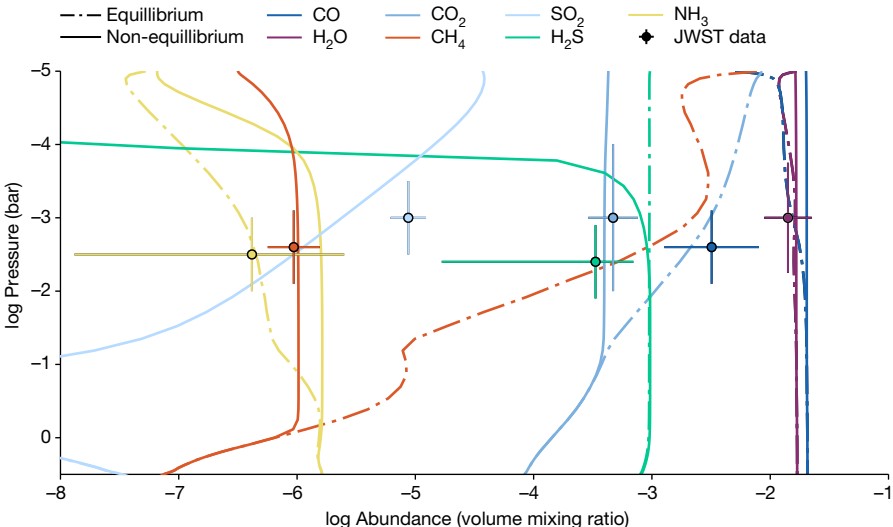

**Fig. 3 | Model interpretation.** ATMO non-equilibrium chemistry model with vertical mixing and photochemistry. The best-fit non-equilibrium chemistry model (solid lines) and the abundance profiles in equilibrium (dot-dashed lines) are shown. The retrieved JWST abundances are shown (data points) with the model grid indicating that the planet has hot interior temperatures ($T_{int} = 458 \pm 38$ K), a super-solar metallicity ($Z/Z_\odot = 43 \pm 8$) with vigorous vertical mixing ($K_{zz} = 10^{11.6 \pm 0.1}$ cm$^2$ s$^{-1}$).

giant planets with several proposed mechanisms, including tidal heating[27], downward transport of kinetic energy[28], enhanced opacities[29], ongoing layered convection[30], ohmic dissipation[31] and, more recently, the advection of potential temperature[32,33]. The potential temperature mechanism has been recently demonstrated in the hot Jupiter WASP-76 b using first-principle radiative GCM simulations[34]. The hot interior temperature can explain the higher vertical mixing rates, as both hotter temperatures and lower surface gravities are expected to increase $K_{zz}$. These high vertical mixing rates with super-solar metallicities and hot intrinsic temperatures also explain the cloud properties observed, in which silicate cloud particles were found with JWST/MIRI[4]. For planets such as WASP-107b with equilibrium temperatures near 770 K, the condensation of silicate clouds should be occurring at high unobservable pressures if $T_{int}$ was low (about 100 K) (ref. 8). However, with a high $T_{int}$ of 460 K, the silicate cloud base is moved to about bar pressures (Extended Data Fig. 6) and the high $K_{zz}$ values found here are more than sufficient to aloft the cloud particles to mbar pressures in which they are observed in transmission.

The mass, radius and $T_{int}$ alone constrain the overall bulk metallicity of the planet to be $Z_p = 63.5^{+10.4}_{-8.5}$%. A precise atmospheric metallicity and $T_{int}$ enable us to place better constraints on the overall metal content of the planet and estimate the core mass of the planet (Methods). So far, no gas giant exoplanet has had the presence of a core significantly detected, with upper limits reported for HAT-P-13 b (ref. 35) and WASP-107b (ref. 10). Any metals seen in the bulk but not in the atmosphere must be hidden in the interior in a core or composition layers. Assuming a uniform composition core, an isothermal 50:50% mixture of rock and water, gives us $M_c = 11.5^{+3.0}_{-3.6}M_\oplus$—which is one-third of the mass of the planet and the first statistically significant core detection for a giant exoplanet. This value is much higher than the previous estimates limiting the core to less than $4.6M_\oplus$ (ref. 10), which had assumed the planet followed a standard cooling track with no tidal heating. These low core-mass values were in tension with standard core-accretion models[11] that do not predict such a light core could accrete the massive H/He envelope observed. By contrast, our core-mass estimate matches the approximately $10M_\oplus$ prediction needed to elicit runaway gas accretion within the lifetime of the disks. It is worth noting that the structure of the interior may not be exactly an envelope-on-core model, but may be more like Neptune and Uranus with a rocky core and an icy water mantle[36] or the core may be diffuse and/or similar to that of Jupiter[37,38]. The core-mass result should be understood as the total excess non-H/He heavy elements in the interior, irrespective of how exactly it is structured. With the proof-of-concept demonstrated here, using $CH_4$ depletion as a thermometer of the deep atmosphere for other gas giant exoplanets could help provide constraints at the population level about how planetary cores might differ with planet or host-star mass and metallicities.

WASP-107b is one of the lowest-density planets known, which has led to speculation that the planet has formed substantially different from the solar system, with dust-free accretion or in situ scenarios explored[10]. However, the important bulk properties found here line up well with the planets of the solar system (SS) indicating a similar core-accretion scenario. In particular, with a metallicity of $43 \pm 8 \times$ solar, WASP-107b is close to the mass-metallicity trend of the gas giant SS planets[39]. Moreover, with an estimated core mass of $11.48^{+3.0}_{-3.6}M_\oplus$, WASP-107b is also intermediate between Neptune and Saturn, falling reasonably along their lines of formation[40,41]. The main difference for WASP-107b seems to be its much hotter interior, resulting in an extremely puffy planet.

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

## Methods

### Data reduction

A transit of WASP-107b was observed on 23 June 2023 with JWST for 6.5 h using the NIRSpec instrument with the G395H grating. This setup gives a wavelength range from 2.7 μm to 5.18 μm with the corresponding resolving power ranging between 1,828 and 3,600. The instrument used the NRSRAPID readout pattern and SUB2048 subarray with 20 groups per integration, resulting in 1,230 integrations over the 6.5-h observation.

### FIREFLy

We reduce the data using the Fast Infrared Exoplanet Fitting Lightcurve (FIREFLy)[17,42] reduction suite, which starts with a customized reduction using the STScI pipeline and the uncalibrated images. Our custom reduction includes $1/f$ destriping at the group level before the ramp is fit. The jump-step and dark-current stages of the STScI pipeline were skipped. We then use the custom-run pipeline two-dimensional (2D) images after the gain scale step and perform customized cleaning of bad pixels, cosmic rays and hot pixels. Using cross-correlation, we measured the positional shift of the spectral trace across the detector and performed shift stabilization of the images with flux-conserving interpolation. This procedure has been found to reduce the amplitude of position-dependent trends[17,42], as the underlying stellar spectra do not shift in wavelength across a pixel through the time series. We note that nearly identical results can be found if shift stabilization of the data is not performed and the extraction window is moved instead. Further position-dependent trends can be expected from intrapixel sensitivity variations[13], although their magnitude on-sky has been very small as JWST generally has excellent pointing with the detector shifts found in this dataset to be of the order of about 1/1,000 of a pixel. We optimized the width of our flux-extraction aperture to minimize the standard deviation of the white-light curve photometry and extracted the spectrophotometric time series. The 2D extracted spectral time series can be seen in Extended Data Fig. 1. We extracted the transmission spectra with a range of different wavelength bin sizes, reporting an extraction with a resolution near $R = 950$.

We fit the transit light curves using a quadratic function to model stellar limb darkening given as

$$\frac{I(\mu)}{I(1)} = 1 - a(1-\mu) - b(1-\mu)^2, \qquad (1)$$

where $I(1)$ is the intensity at the centre of the stellar disk, $\mu = \cos(\theta)$, where $\theta$ is the angle between the line of sight and the emergent intensity, and $a$ and $b$ are the limb-darkening coefficients. In practice, we remap $a$ and $b$ and fit instead for $q_1$ and $q_2$, which is shown to be less degenerate[43]. The resulting limb-darkening coefficients can be seen in Extended Data Fig. 3 with the best-fit orbital parameters given in Extended Data Table 1. Model limb-darkening coefficients calculated from three-dimensional models[44] capture the wavelength dependence of limb darkening well, but we find an overall offset in the coefficients of −0.0378 and +0.175 for $q_1$ and $q_2$, respectively. We applied these offsets to the wavelength-dependent model limb-darkening coefficients and re-fit the transmission spectra with fixed limb darkening. This iterative procedure reduces the bin-to-bin scatter of the transmission spectra as the limb darkening is fixed, but allows for overall differences between the transit data and stellar model limb darkening. We bin the spectro-photometry into 576 wavelength bins, providing a spectral resolution of $R \approx 950$ for the planet.

### Stellar spot crossing

A stellar spot-crossing event is observed in our transit light curve data, appearing in both NRS1 and NRS2 shortly after mid-transit, which we decontaminated from the light curves and the transmission spectrum.

The host star is known to be modestly active[12], with optical photometric modulations of around 0.4% over a period of about 17 days. We modelled this spot empirically, first fitting the non-spotted region with a transit light curve and using the overall residual in the spotted region to define the photometric shape of the spot (Extended Data Fig. 2). A Gaussian filter was then applied to the spot shape and subsequently used to model the full transit light curves, fitting for a scaling factor. As the spot is low in amplitude and covers only a small portion of the data, our overall transmission spectrum is insensitive to the occulted spot. We found equivalent results between masking the region for all the transit light curve fits and fitting for the spot shape in the light curves.

### TEATRO

An independent data reduction was performed using the Transiting Exoplanet Atmosphere Tool for Reduction of Observations (TEATRO) (https://github.com/ncrouzet/TEATRO) pipeline.

The data were processed using the jwst calibration software package v.1.10.2, CRDS v.11.16.19, CRDS context 1147 (refs. 45,46), starting from the 'uncal' files. For stage 1, a jump rejection threshold of 6 was used and the 'jump.flag_4_neighbors' parameter was turned off. For stage 2, the flat field and photometric calibration steps were skipped. Bad pixels and missing pixels were then corrected in each integration. The centre of the spectral trace in the spatial direction was computed by fitting a Gaussian function to each column and fitting a third-order polynomial to their maxima. For each integration, the background was computed on a column-by-column basis using pixels above and below the spectral trace (or only one of them at the short- and long-wavelength edges of NRS2). An aperture half-width of 2.73 and 3 pixels and background regions starting 6.5 and 6 pixels further away were used for nrs1 and nrs2, respectively. The background subtraction and spectral extraction were performed using the 'extract_1d' routine with the corresponding polynomial coefficients as parameters. The white-light flux was computed using the 'white_light' step routine. For the spectroscopic analysis, we binned the spectra in 86 and 136 wavelength bins of 0.01 μm width, covering the 2.86–3.72 μm and 3.82–5.18 μm ranges with nrs1 and nrs2, respectively. The light curves were normalized by the out-of-transit flux and a five-iteration 3.5 sigma-clipping was applied to remove the outliers.

Light-curve fits were performed using the exoplanet[47,48] package with a quadratically limb-darkened transit light-curve model and a second-order polynomial to account for long-term trends. A fit to the white-light curve obtained from NRS1 only was first performed to derive system parameters (mid-transit time, impact parameter, planet-to-star radius ratio and stellar density). These parameters were then fixed for the spectroscopic light-curve fits, leaving only the radius ratio and polynomial trend as free parameters. Limb-darkening coefficients were obtained from ExoTETHyS[49]; they were free for the white-light curve fit and fixed for the spectroscopic light-curve fits. The median of the posterior distribution of the radius ratio was used to derive the transit depth in each wavelength bin. The uncertainties were computed by summing quadratically the standard deviations of the in- and out-of-transit parts of the light curve divided by the square root of their respective number of points (this gave a better estimate than the standard deviation of the posterior distributions).

Comparing the TEATRO reduction to FIREFLy, we find good consistency with the spectra agreeing at the point-to-point level at better than $1\sigma$ for both NRS1 and NRS2.

### Resolution-linked bias

We have adjusted the retrieved abundance to account for the resolution-linked bias (RLB) effect[50], in which planetary absorption signatures are diluted within stellar absorption lines. For G395H data, this, in particular, dilutes the transmission spectral signatures of CO, as the molecule is found in the atmosphere of both the star and planet. We followed

the method in ref. 42, estimating the magnitude of the effect using high-resolution models. Compared with WASP-39A, WASP-107A is a cooler later-type star that enhances the RLB effect as the stellar CO is stronger. However, within the CO lines, the atmosphere of the planet is cloudier than WASP-39b, which reduces the effect. In total, we find that the RLB reduces the inferred CO abundance by 0.2 dex, which is about half of the uncertainty in the CO abundance.

## Atmospheric models

**ATMO setup.** We use ATMO, a one-dimensional (1D)–2D radiative–convective equilibrium model for planetary atmospheres to generate models of WASP-107b. More comprehensive descriptions of the model can be found in refs. 18,32,51–54. The ATMO model is used here as both a forward physical model in radiative and chemical (dis-)equilibrium and a retrieval model in which the abundances and temperature–pressure ($T$–$P$) profile are let free to fit the data. By comparing the retrieved and forward model abundances within a single model suite, model-to-model systematics from such sources as differences in opacities can be avoided. ATMO has been validated against the publicly available MET office radiative transfer code SOCRATES[55] and has been benchmarked against Exo-REM[56] and petitCODE[57] in the context of JWST observations[58].

ATMO solves the radiative transfer equation with isotropic scattering in 1D plane-parallel geometry for the irradiation and thermal emission, finding a pressure–temperature ($P$–$T$) profile that satisfies hydrostatic equilibrium and conservation of energy and is also self-consistent with the atmospheric chemistry and opacities, given a set of elemental abundances. The total opacity of the gas mixture is computed using the correlated-$k$ approximation using the random overlap method with resorting and rebinning[51,59]. The $k$-coefficients are calculated 'on the fly' for each atmospheric layer, spectral band and iteration such that the derived opacities are physically self-consistent with the $T$–$P$ profile and chemical composition. ATMO can also be used as a full line-by-line code at high spectral resolution, although that is not used here as it is too computationally heavy for spectral retrieval purposes. The spectrally active species currently include $H_2$–$H_2$ and $H_2$–He collision-induced absorption (CIA) opacities, as well as $H_2O$, $CO_2$, CO, $CH_4$, $NH_3$, Na, K, Li, Rb, Cs, $TiO$, VO, Fe, FeH, CrH, $PH_3$, HCN, $C_2H_2$, $H_2S$, $SO_2$ and H⁻ (see refs. 54,60 for full description). Multi-gas Rayleigh scattering contributions from the different species are also included.

The chemical abundances in equilibrium are determined by minimizing the Gibbs free energy following the method in ref. 61 and using the thermochemical data from ref. 62. Solar elemental abundances are set from refs. 63,64, with the chemistry fully flexible for any mix of input elemental abundances. This method also enables the depletion of gas phase species because of condensation as well as thermal ionization and dissociation. ATMO has several different chemical schemes that can be chosen and, by default, calculates the abundances for 175 neutral, 9 ionic and 93 condensate species. Condensation can be treated locally or with rainout. Rainout is treated by calculating the chemistry at the highest pressure initially, then following the $T$–$P$ profile towards lower pressures. Condensed elements are removed locally, as well as for all lower pressures[65], allowing for opacity changes that can alter the radiative–convective balance and $P$–$T$ profile.

For non-equilibrium C–H–N–O chemistry, ATMO uses a chemical kinetics scheme from ref. 66 as described in ref. 18. As done in ref. 7 to model the S photochemistry of WASP-39b with ATMO, we used the thermochemical network from ref. 67 along with the photochemical scheme from ref. 68, including 71 photolysis reactions of $H_2S$, $S_2$, $SO_2$, SO, $SO_2$, $CH_3SH$, SH, $H_2SO$ and COS.

ATMO includes a relatively simple treatment of clouds and hazes[54] and does not consider the distribution of aerosol particles. An aerosol 'haze' scattering is implemented as enhanced Rayleigh-like scattering[69], presented as

$$\sigma(\lambda)_{haze} = \delta_{haze}\sigma_0(\lambda/\lambda_0)^{-\alpha_{haze}}, \tag{2}$$

where $\sigma(\lambda)$ is the total scattering cross-section of the material, $\delta_{haze}$ is an empirical enhancement factor, $\sigma_0$ is the scattering cross-section of molecular hydrogen at 0.35 μm and $\alpha_{haze}$ is a factor determining the wavelength dependence with $\alpha_{haze} = 4$ corresponding to Rayleigh scattering. Condensate 'cloud' absorption is assumed to have a grey wavelength dependence and is calculated as

$$\kappa(\lambda)_{cloud} = \delta_{cloud}\kappa_{H_2}, \tag{3}$$

where $\kappa(\lambda)_{cloud}$ is the 'cloud' absorption opacity, $\delta_{cloud}$ is an empirical factor governing the strength of the grey scattering and $\kappa_{H_2}$ is the scattering opacity due to $H_2$ at 0.35 μm. $\sigma(\lambda)_{haze}$ and $\kappa(\lambda)_{cloud}$ are added to the total gaseous scattering and absorption, respectively.

For spectra retrievals, we coupled the forward ATMO model to a nested sampling statistical algorithm to marginalize the posterior distribution and measure the model evidence[70–72]. The retrieval aspects of ATMO were developed to fit transit and eclipse data, with results published for several transiting planets[3,73–77]. A main difference compared with the forward models is that the retrieval does not converge the $T$–$P$ profile to radiative–convective equilibrium, but rather parameterizes the profile that is then fit against the planetary spectrum. With the retrieval model, the $k$-coefficients are still calculated 'on the fly' in the same way as described in the forward model (and equilibrium chemistry if specified) for every likelihood evaluation step to maximize accuracy and consistency. To parameterize the $T$–$P$ profile, we use the three-channel (two optical, one infrared) analytic radiative equilibrium model as described in ref. 78, which is based on the derivations in ref. 79.

**ATMO WASP-107b retrievals.** For the NIRSpec WASP-107b data, we found the flexible parameterized $P$–$T$ profile simply fit to a simple isotherm, so we instead fit for a single isothermal temperature for the atmosphere. Given the atmosphere of the planet is far out of equilibrium, we fit a constant VMR for each molecule. Apart from the cloud, we included the spectrally active species of $H_2$–$H_2$, $H_2$–He (CIA) opacities, $H_2O$ (ref. 80), $CO_2$ (ref. 81), CO (ref. 82), $CH_4$ (ref. 83), $NH_3$ (ref. 84), $H_2S$ (ref. 85) and $SO_2$ (ref. 86).

We searched the G395H data for $H_2S$ and can marginally improve the model fits to the data, but the inclusion of the model is not supported with confidence by the Bayesian evidence. Considerable opacity from aerosols is also needed to fit the spectrum, which is in line with previous HST results in which $H_2O$ and He features are observed between 0.9 μm and 1.6 μm peaking above the clouds[2,75]. We fit the data with a grey cloud and a scattering haze but found only a grey cloud was needed to fit the data (Extended Data Table 3). Moreover, the choice of cloud parameterization did not have a strong influence on the retrieved abundances. Fitting the grey cloud either with a single uniform opacity or for a cloud-top pressure also gave similar results. A narrow feature of $NH_3$ is potentially found in the data at 3.0 microns, but similar to $H_2S$, its identification is not supported with high confidence by the Bayesian evidence. $NH_3$ has been marginally detected at longer wavelengths with JWST/MIRI, indicating that the molecule could be confidently identified with a dedicated multi-wavelength study.

The retrieval constraints are given in Extended Data Table 2. CO is found to have abundances of $10^{-2.7\pm0.4}$, which is slightly low relative to 40× solar values derived from $H_2O$. Although low, the uncertainty on the CO abundance is large as the feature is subtle in the data and $H_2O$ and clouds help mask the signature. A confounding factor regarding CO is an RLB effect[17,50], where CO present in the stellar spectra dilutes the strong molecular line cores in the planetary transmission spectrum. This does not affect $H_2O$, $CH_4$ or $CO_2$ as those molecules are not present in the stellar photosphere. In this work, we applied a correction for the CO abundance to account for the RLB effect.

Overall, we find a good fit to the data with a $\chi_\nu^2 = 1.1$ for 566 degrees of freedom and 10 free parameters. Our retrieval results are given in Extended Data Table 2 and Extended Data Fig. 4. To estimate the detection confidence of each molecule identified, we re-ran the retrieval leaving out the molecule in question and computed the detection significance using the Bayesian evidence between the models with and without the molecule. We report the results in which $H_2S$ and $NH_3$ are included in the model, although the G395H data are not sufficient to detect either species confidently. We report the abundances and detection significances from the ATMO results, as they can be directly used to compare against the non-equilibrium models presented in the main text.

**ATMO WASP-107b forward non-equilibrium chemistry models.** It is computationally infeasible to currently run the ATMO retrieval with the full non-equilibrium photochemistry self-consistently computed at every model evaluation 'on the fly'. Instead, we adopt a two-step grid-retrieval approach, first computing a grid of non-equilibrium chemistry values that are then fit against the retrieved VMR abundances. As we use the same ATMO model setup for both the forward and retrieval modelling, the abundances computed between them can be self-consistently compared.

Our use of a 1D atmospheric model to interpret the non-equilibrium chemistry has several assumptions. The first is that the atmosphere can be considered 1D and is independent of both latitude and longitude. Hot and ultra-hot Jupiters show large day–night temperature gradients[87]. Spitzer phase curves show these differences decrease at lower $T_{eq}$ (ref. 88). Thus, the latitude and longitude temperature differences for warm about 750 K planets such as WASP-107b are expected to be modest. Another important assumption is that the parameters of $T_{int}$ and $K_{zz}$ are constant and assumed to be uniform with depth. Our model contains a single convective region (Extended Data Fig. 6), although the possibility of a separate low-pressure convective zone has been studied[89]. Multiple convective regions would affect our interpretation of $T_{int}$ and the link between atmospheric and interior constraints. Furthermore, our modelling also assumes a single $K_{zz}$ value that is constant with altitude. Theoretical studies have predicted that $K_{zz}$ increases with altitude[20]. Higher $K_{zz}$ values at higher altitudes would not greatly affect our results, as the quench pressures for the molecular features probed here are similar (Extended Data Fig. 6). However, if multiple convective zones are present, the use of a single $K_{zz}$ value may have to be re-visited.

To estimate the parameter space needed for the grid, we first used a chemical equilibrium model[90] and assumed a single quench pressure to estimate the posterior distribution of temperature, pressure, metallicity and C/O ratio parameters that are consistent with the retrieved abundances (Extended Data Fig. 5). This equilibrium model indicated metallicities near 50× solar, C/O ratios near 0.5 and temperatures near 1,000 K at pressures just below a bar are needed to fit the observed molecular abundances. These temperatures are reached at 1 bar only if the planet has an intrinsic temperature greater than about 300 K with super-solar metallicities (Extended Data Fig. 6). Intrinsic temperatures lower than 300 K are ruled out as $CH_4$ becomes too abundant in the deep atmosphere, such that no amount of mixing could sufficiently deplete the molecule. We also estimate a limit on $K_{zz}$ to be less than $10^{13.5}$ cm$^2$ s$^{-1}$ based on requiring velocities to be less than the local sound speed. The presence of clouds near millibar pressures also places a lower limit on $K_{zz}$ of about $10^9$ cm$^2$ s$^{-1}$, as significant vertical mixing is needed to keep the aerosol particles aloft[20]. We input $P$–$T$ profiles in radiative–convective equilibrium, using intrinsic temperatures ($T_{int}$) of 300–500 K in steps of 50 K along with a $T_{int}$ of 425 K. With these $T$–$P$ profiles, we computed the non-equilibrium chemistry as described in ref. 18, using metallicities ($Z$) between 20× solar and 50× solar in steps of 5×, and 10 log[$K_{zz}$ (cm$^2$ s$^{-1}$)] values ranging from 11 to 13. A K6V star was used for our input to the photochemical model[91]. Each non-equilibrium model was integrated to $1 \times 10^8$ s, which

we found was sufficient such that the abundances did not evolve further. We assumed a solar C/O ratio, which is also consistent with WASP-107A from ref. 92 which found values of C/O = 0.5 ± 0.1. We find the C/O ratio does not differ substantially from the solar or host-star value (Extended Data Fig. 5), and we subsequently ran low (0.1 and 0.2) and high (0.7) non-equilibrium chemistry cases that did not improve the fits.

For this grid of models, we then calculated the $\chi^2$ for each grid model, using the retrieved abundances for $H_2O$, CO, $CO_2$, $CH_4$ and $SO_2$ in Extended Data Table 2. We did not use the $H_2S$ or $NH_3$ abundances, as they are not fully supported by the data and the errors in any case are large. The best-fit model has a good $\chi^2$ of 4.1 when fitting the five abundances with three grid parameters (Fig. 3 and Extended Data Fig. 7). The grid-posterior can be seen in Extended Data Fig. 7, in which each model was given a weight proportional to its probability, $P$, calculated as $P = e^{-\chi^2/2}$. From the marginalized grid-posterior, we fit a Gaussian to the distributions and report the fitted $Z/Z_\odot$, $T_{int}$ and $K_{zz}$ values in Extended Data Table 2.

We find the molecular constraints highly constraining for both $T_{int}$ and $K_{zz}$. Lower vertical mixing rates, generally, require higher intrinsic temperatures to deplete $CH_4$ to similar values. However, if $T_{int}$ becomes too high, it over-depletes $CH_4$ relative to $CO_2$. Moreover, $SO_2$ helps provide an upper bound on $K_{zz}$ as $SO_2$ is sensitive to both the photochemistry, needed to produce the species at low pressures, and vertical mixing. Higher values of $K_{zz}$ mix the species from deeper pressures at which the abundances are lower, whereas lower $K_{zz}$ values allow the photochemistry to build up the species near 0.1 mbar pressures.

**NEMESIS free retrievals.** NEMESIS is a free-chemistry radiative transfer and retrieval code, initially developed for solar system applications[93]. It couples a correlated-k[59] radiative transfer scheme with the PyMultiNest[70–72,94] Nested Sampling algorithm[95]. It has been extensively applied to spectra of hot Jupiter exoplanets[96–98].

The retrievals for WASP-107b incorporate abundances of the gases $H_2O$ (line data[99]), $CO_2$ (ref. 100), CO (ref. 101), $CH_4$ (ref. 102), $SO_2$ (ref. 86), $NH_3$ (ref. 103), HCN (ref. 104) and $H_2S$ (ref. 105). For each of the gas abundances, we specify a log-uniform prior spanning $10^{-12}$–$10^{-0.5}$. The rest of the atmosphere is composed of $H_2$ and He with a ratio of 0.8547:0.1453. CIA for $H_2$ and He is taken from refs. 106,107.

Other retrieved properties are an isothermal temperature with a uniform prior and a range of 300–1,000 K, and a grey cloud-top pressure with a log-uniform prior and a range of $10^{-8}$–100 bars. We also retrieve the planetary radius at a reference pressure of 100 bars. The retrieved abundances are given in Extended Data Table 2. Additional retrievals were run that had a more flexible $T$–$P$ profile along with retrievals with patchy clouds and enstatite clouds, although they were not favoured statistically over the simpler model.

**Interior structure models.** The constraints we found for the intrinsic temperature of WASP-107b have important implications for the interior structure of the planet. To investigate these, we use the interior structure models of ref. 39, revised to use the H/He EOS[108] and parameterize the amount of metal in the core versus mixed into the envelope. As in ref. 39, we assume the metal is a 50–50 mixture of rock and ice.

To match our models to WASP-107b, we use a Bayesian retrieval approach similar to the one used in ref. 109. However, our measurement of $T_{int}$ determines the thermal state of the plane rather than thermal evolution. The parameters of the statistical model were, therefore, the mass of the planet $M$, the metallicity of the envelope $Z_e$ (a mass fraction), the core mass $M_c$ and the specific entropy of the envelope $s$. From these, our forward model calculates the radius and $T_{int}$ of the planet. The model was constrained by four normally distributed observations: the mass, the radius, the atmospheric metallicity and $T_{int}$. These are constraining enough that we can choose very uninformative priors to keep the parameters inside model bounds: a uniform mass prior from

0.01 to 30 $M_J$, a uniform $Z_{env}$ from 0 to 1, a uniform core mass from 0 to $M_J$ (thus conditional on $M$ but still proper) and a uniform entropy prior from 5 to 11 $k_b$ per baryon. We sampled from the posterior using the Metropolis–Hastings Markov chain Monte Carlo algorithm and verified convergence with the auto-correlation plots and the Gelman–Rubin statistic[110].

Our interior model retrievals (Extended Data Fig. 8) find a planet that contains similar amounts of H/He compared with heavier elements, with $Z_p = 0.608 \pm 0.046$. This seems reasonable for a planet of this mass: it accreted a substantial amount of material from the disk, but never reached the runaway gas accretion stage. However, this is in contrast to the previous estimates of the composition of the planet, which saw it as having substantially larger quantities of H/He: that is, more than 85% H/He from ref. 10. This is because our measured $T_{int}$ is much hotter than a standard thermal evolution model would suggest; we would expect the planet to have cooled below our $2\sigma$ lower bound of 300 K within tens of Myr of formation. Instead, it is clear that the planet is being inflated by an extra heat source despite being too cool to experience hot Jupiter-type inflation[111]. Instead, a mechanism such as tidal heating[112–114], previously suspected to be operating on WASP-107b (refs. 10,115), is warming the interior. These models propose that the eccentricity of WASP-107b is excited by WASP-107 c, but then damped out through tidal interaction with the star; the resulting energy is deposited in the interior of the planet. Our large measured $T_{int}$ is evidence that this process could be occurring, with a precise eccentricity further capable of constraining the mechanism.

Another interesting result of our interior structure models is that it puts constraints on the mass of the core of the planet. The mass, radius and $T_{int}$ already constrain the overall metallicity of the planet (more metal means a smaller planet) to $Z_p = 0.635^{+0.104}_{-0.085}$, but we also have a measurement of the atmospheric metallicity of the planet. This enables us to make a modestly better constraint on $Z$ (see previous paragraph); more importantly, metal seen in the bulk but not in the atmosphere must be hidden in the interior in a core or composition layers (see ref. 109 for further discussion). Assuming a uniform composition core gives us $M_c = 11.5^{+3.01}_{-3.58} M_\oplus$. This range excludes zero, meaning that we have detected the presence of a core. The exact structure of the interior may not be exactly an envelope-on-core model, but may be more like Neptune and Uranus with a rocky core, water envelope, than a H/He envelope[36]. Alternatively, the core may be diffuse and/or layered as the core of Jupiter is thought to be[37,38]. Nevertheless, our result that a core exists is not affected by these possibilities: we cannot propose a layered core without a core in the first place.

## Data availability

The data used in this paper are associated with the JWST programme GO 1224 (PI Birkmann) and are publicly available from the Mikulski Archive for Space Telescopes (https://mast.stsci.edu) from 23 June 2024. The data products in Figs. 1, 2 and 3 are available at Zenodo (https://doi.org/10.5281/zenodo.10891400) (ref. 116). Source data are provided with this paper.

## Code availability

The codes used in this publication to extract, reduce and analyse the data are as follows: STScI JWST Calibration pipeline (https://github.com/spacetelescope/jwst) and FIREFLy[17,42]. Moreover, these codes made use ExoTiC-LD[117] (https://exotic-ld.readthedocs.io/en/latest/) and Emcee (https://emcee.readthedocs.io/en/stable/)[118], which use the Python libraries scipy[119], numpy[120], astropy[121] and matplotlib[122]. Model and retrievals were generated using ATMO, a proprietary code extensively described in refs. 18,32,51–54 and NEMESIS[93] (https://github.com/nemesiscode/radtrancode).

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

**Acknowledgements** This work is based on the observations made with the NASA/ESA/CSA JWST. The data were obtained from the Mikulski Archive for Space Telescopes at the Space Telescope Science Institute, operated by the Association of Universities for Research in Astronomy, under NASA contract NAS 5-03127 for JWST. E.K.H. Lee is supported by the SNSF Ambizione Fellowship grant (#193448). J.K.B. is supported by UK Research and Innovation via an STFC Ernest Rutherford Fellowship (ST/T004479/1).

**Author contributions** D.K.S. led the data analysis and modelling. Z.R., N.E. and N.C. provided data analysis and input. J.K.B., Z.R. provided atmospheric models. D.P.T. provided planetary interior models. S.M.B. led the observational setup and execution of the program. R.C.C., N.K.L. and E.K.H.L provided support interpreting the results. C.A.d.O., T.L.B., S.M.B., P.F., G.G., R.M., E.M., B.J.R., M.S., A.G. and J.A.V. contributed to the NIRSpec instrument and GTO team. The manuscript was written by D.K.S. along with D.P.T., J.K.B. and P.T.

**Competing interests** The authors declare no competing interests.

**Additional information**
**Correspondence and requests for materials** should be addressed to David K. Sing.

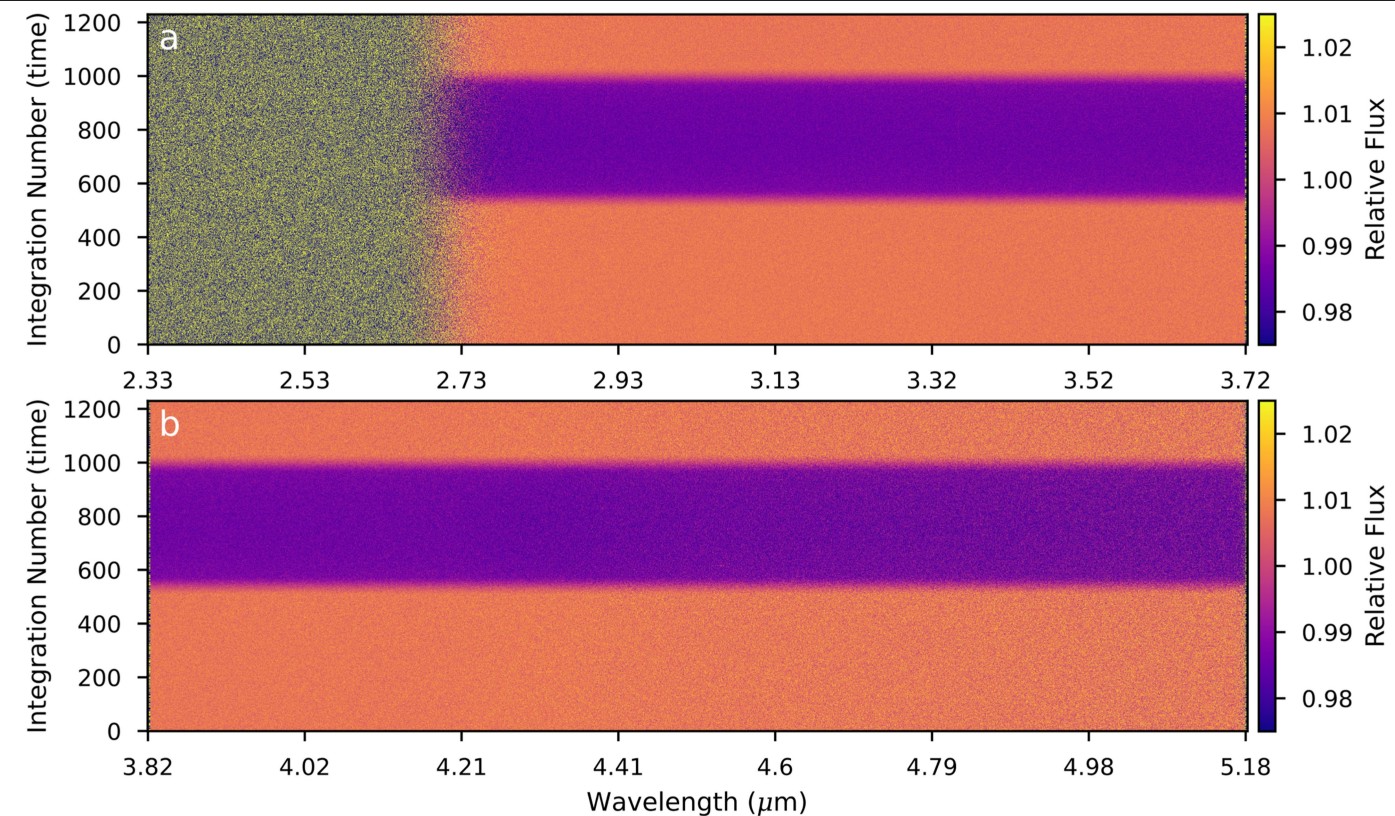

**Extended Data Fig. 1 | FIREFLy transit light curve spectrophotometry.** Shown is the relative flux as a function of wavelength and time for NIRSpec detectors (**a**) NRS1 and (**b**) NRS2.

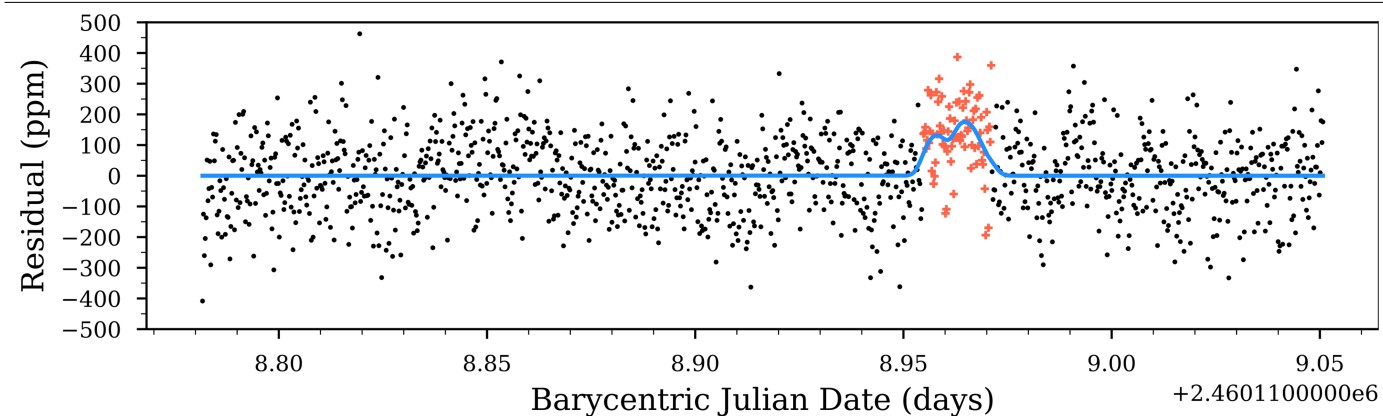

**Extended Data Fig. 2 | Stellar spot-crossing.** Shown is the residual white-light curve photometry from NRS1 when fitting for the non-spotted data (black dots). The suspected occulted spot (red crosses) is shown, with a Gaussian smoothed filter overplotted (blue line).

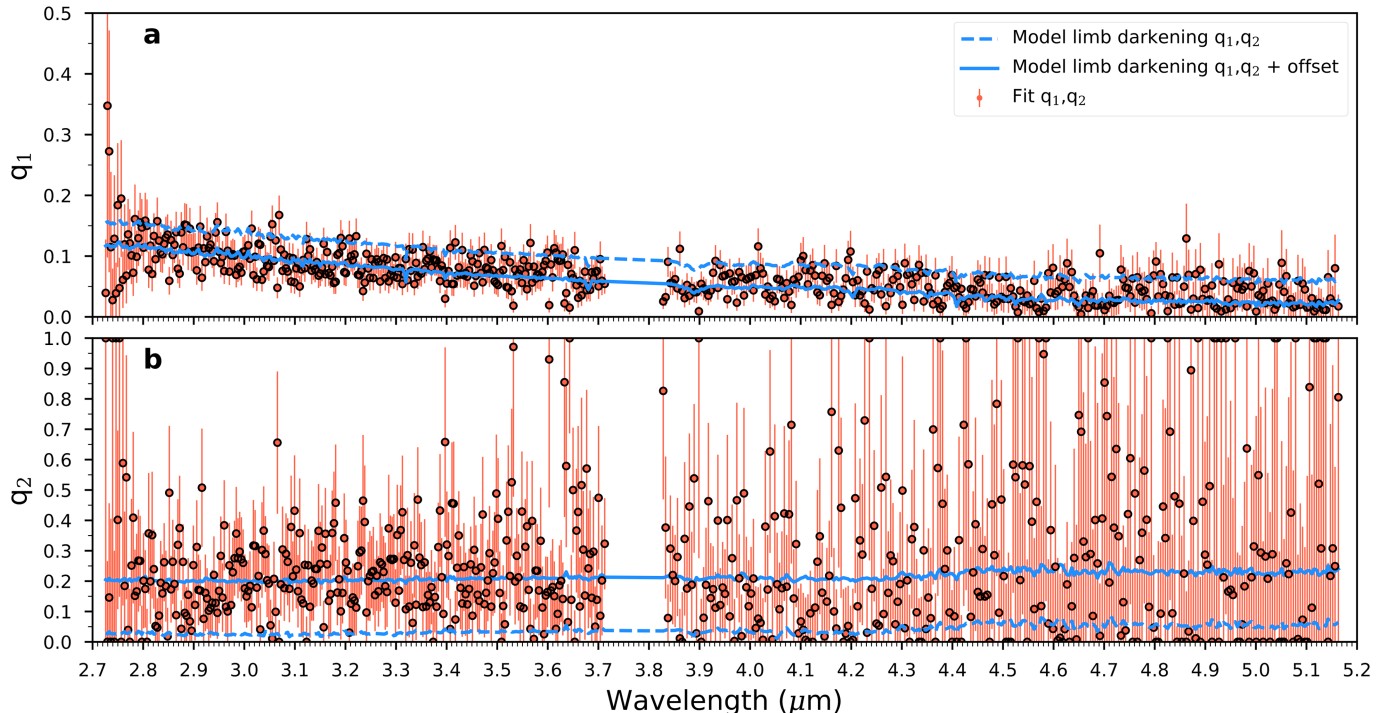

**Extended Data Fig. 3 | Limb-Darkening.** Shown are the resulting stellar limb-darkening coefficients $q_1$ (**a**) and $q_2$ (**b**) derived from the transit light curves using a quadratic law. The limb-darkening coefficients derived from a stellar model are also shown, along with the models with an offset derived from the difference between the data and model.

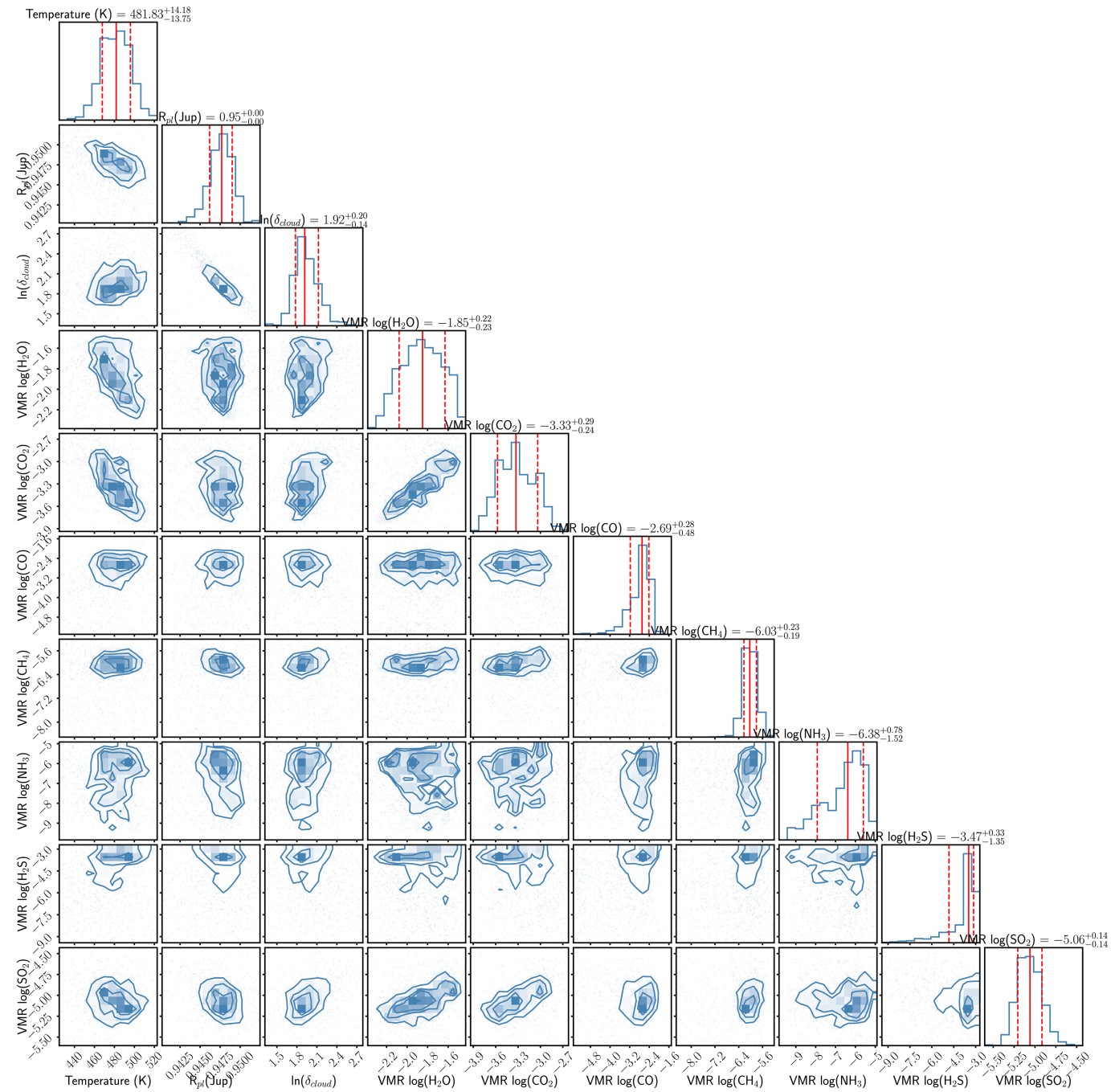

**Extended Data Fig. 4 | WASP-107 b retrieval posteriors.** Shown is the distribution for the ATMO free-retrieval. VMR refers to the Volume Mixing Ratio of the molecular species. 1, 1.5, and 2-$\sigma$ equivalent contours are shown. The 1D histograms show the marginalized parameter median value and 1-$\sigma$ range (red).

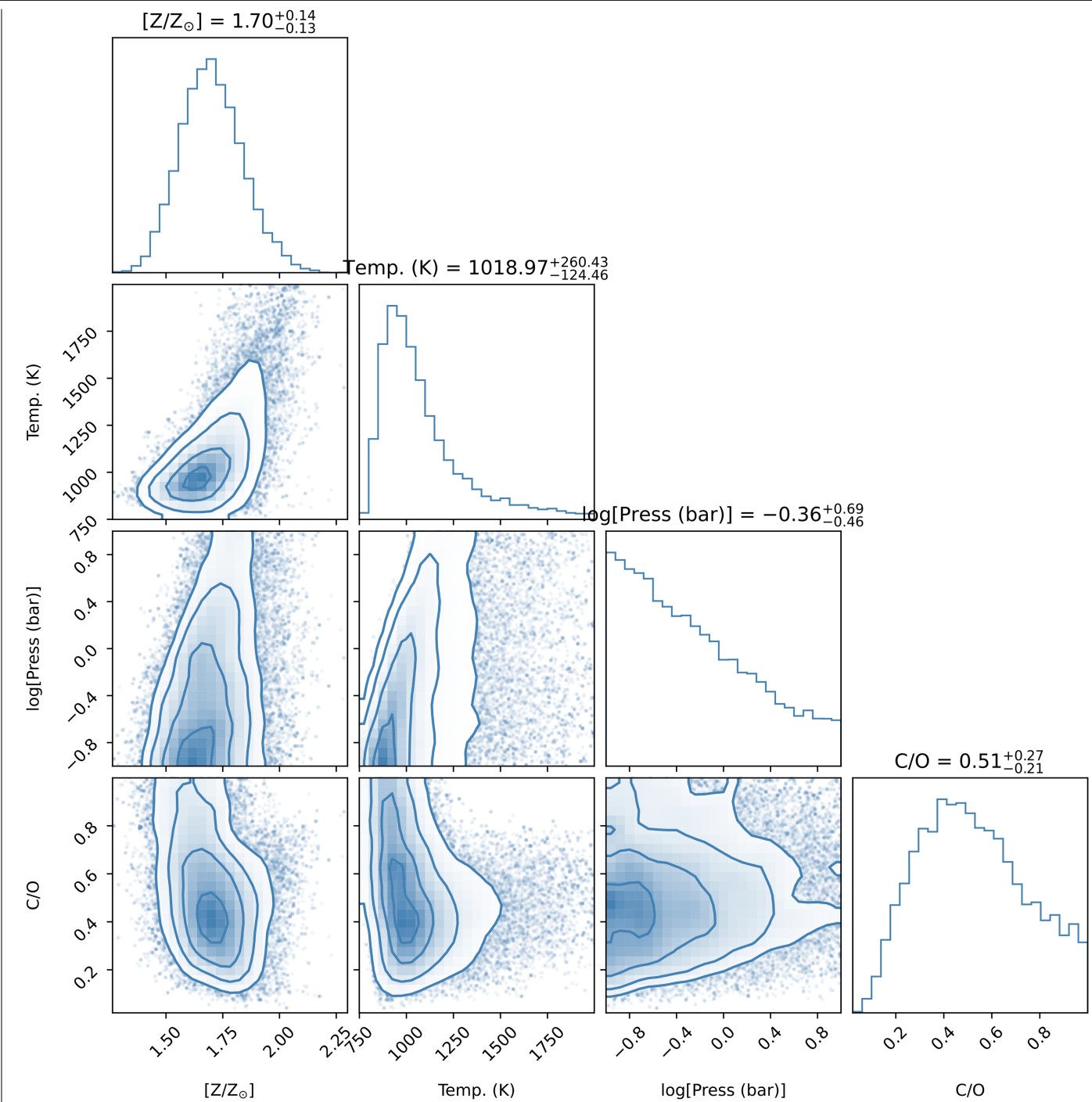

**Extended Data Fig. 5 | Equilibrium chemistry estimation.** Shown is a posterior distribution of metallicity, temperature, pressure and C/O equilibrium chemistry[90] values that are simultaneously compatible with the retrieved abundances of $H_2O$, $CO$, $CO_2$.

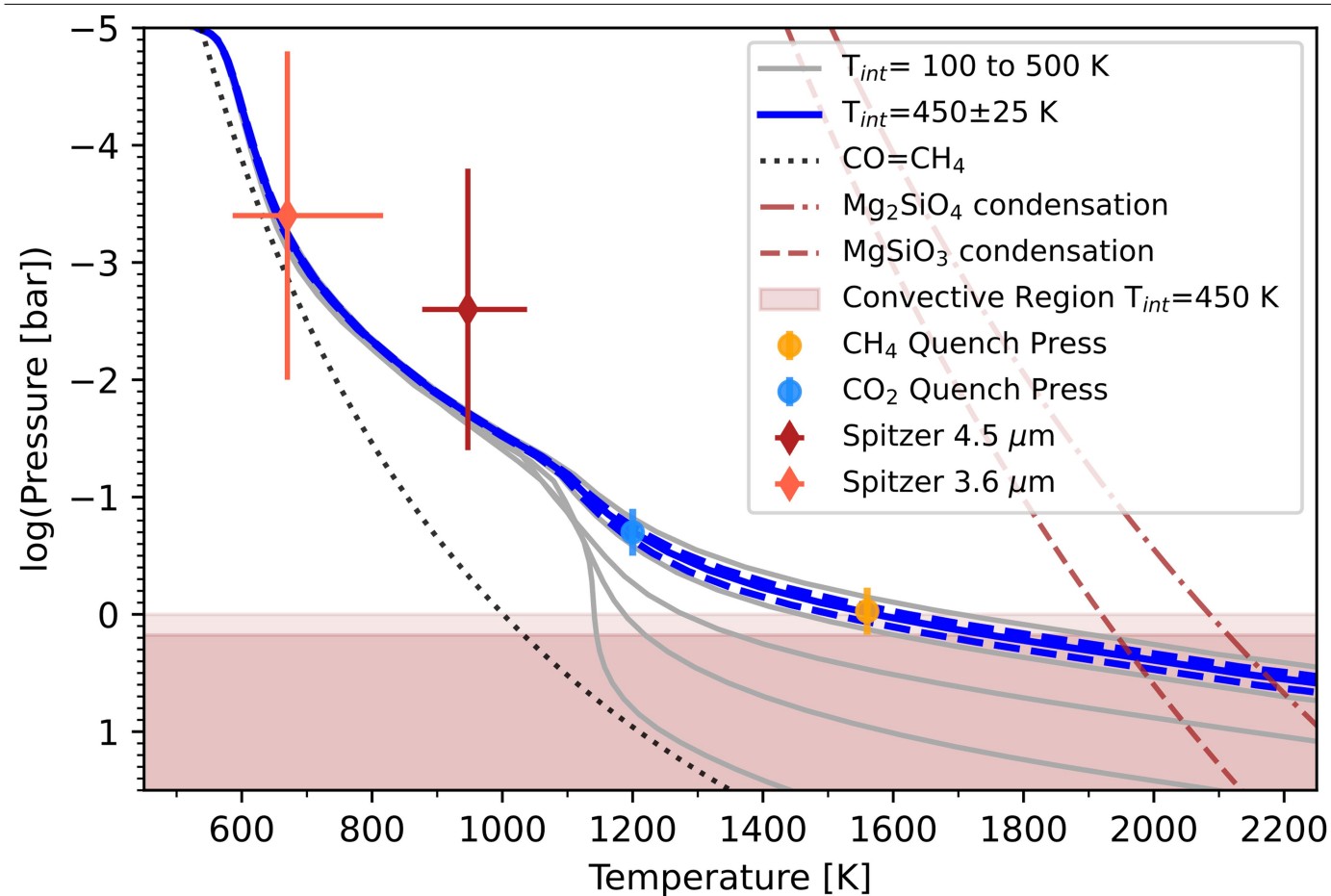

**Extended Data Fig. 6 | Pressure-Temperature Profiles.** Shown are P-T profiles in radiative-convective equilibrium with $T_{int}$ values ranging from 100 to 500 K (grey). The T-P with the best-fit $T_{int}$ is shown (blue), with a shaded region showing where the model is dominated by convection. The quench pressures for $CO_2$ and $CH_4$ are also depicted along with Mg-Si condensation curves (dashed, dot dashed lines). The equilibrium $CH_4$=CO equal-abundance curve is also shown (dotted line), with the $CH_4$ abundance dropping at increased temperatures. The brightness temperatures measured from Spitzer secondary eclipse observations are shown from ref. 123. The corresponding pressures and ranges are derived from the best-fit model contribution function, with the y-axis range encapsulating 80% of the total emitted flux. The Spitzer brightness temperatures are consistent with the best-fitting $T_{int}$ = 460$K$ T-P profile.

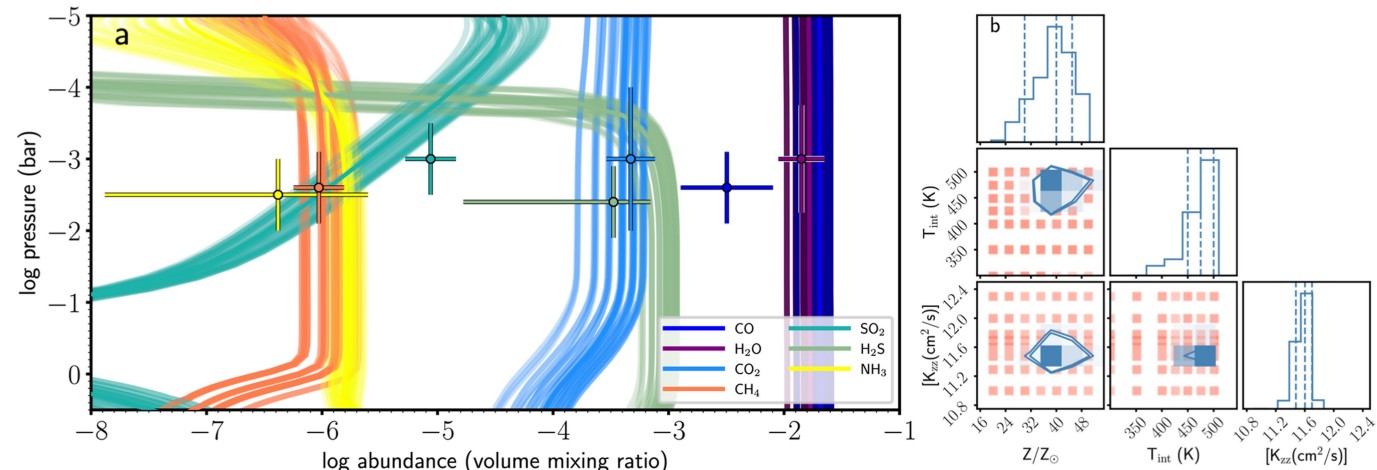

**Extended Data Fig. 7 | WASP-107b forward non-equilibrium chemistry model grid results.** (**a**) Shown are the best-fitting chemical abundances (within 1-$\sigma$) from the non-equilibrium chemistry models along with the retrieved values from the JWST transmission spectrum (datapoints). (**b**) The corner plot depicts the forward model grid points (red squares) along with the constraints in atmospheric metallicity ($Z/Z_\odot$), intrinsic temperature ($T_{int}$), and eddy diffusion coefficient ($K_{zz}$).

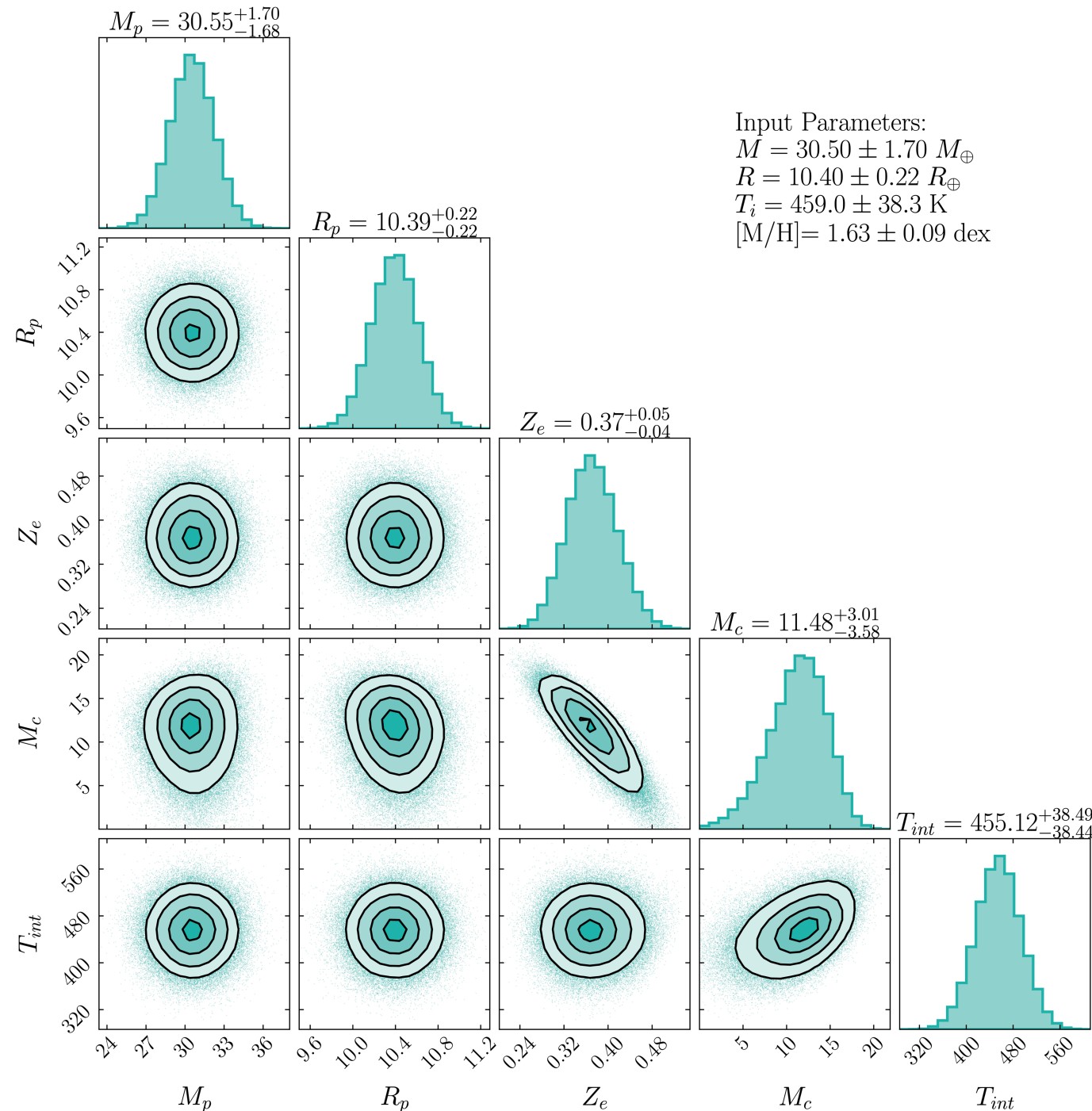

**Extended Data Fig. 8 | Interior structure modeling constraints.** A corner plot of the posterior mass ($M_p$), radius ($R_p$), envelope metallicity (unitless), core mass ($M_\oplus$), and intrinsic temperature (K). The model inputs (from observations) are shown in the upper-right, and the priors were weakly-informative. The overall bulk metallicity is set by $M$, $R$, and $T_{int}$, and can be seen as an arc in $Z_e$-$M_c$ space. Our atmospheric constraint restricts us to a section of this arc; without it, the two parameters would be fully degenerate, running from $M_c = 0$ on one side to $Z_e = 0$ on the other, though the effect on $Z_p$ would be much more limited. The intrinsic temperature is significantly higher than unheated evolution models would produce, and is thus evidence of tidal heating (see text).

**Extended Data Table 1 | Orbital parameters**

| Parameter | Value | Description | Reference |
|:---:|:---:|:---:|:---:|
| $P$ | 5.7214742±0.0000043 | | [124] |
| $T_0$ | 2460118.948861 ± 0.0000063 | Mid-transit time days $(\mathrm{BJD_{TDB}})$ | this work |
| $a/R_\star$ | 18.0923 ± 0.0212 | Scaled semi-major axis | this work |
| $b$ | 0.11650 ± 0.01199 | Transit impact parameter | this work |

Best-fit orbital parameters as measured from the FIREFLy JWST white light curve. Reference 124.

**Extended Data Table 2 | Model Results**

| Parameter | Value & 1-$\sigma$ Uncertainty | Detection ($\sigma$) | Value & 1-$\sigma$ Uncertainty |
|---|---|---|---|
| | ATMO Retrieval | | NEMESIS Retrieval |
| Atmosphere Temperature (K) | $482^{+14}_{-13}$ | | $474^{+17}_{-11}$ |
| $R_{pl}$ ($R_J$) | $0.9477^{+0.0013}_{-0.0015}$ (mbar) | | $0.88\pm0.01$ (100 bar) |
| cloud $\delta_{cloud}$ | $1.91^{+0.20}_{-0.14}$ | | |
| log (cloud top) (bar) | | | $-3.74^{+0.12}_{-0.16}$ |
| VMR log($H_2O$) | $-1.85^{+0.22}_{-0.23}$ | 17 | $-1.72^{+0.26}_{-0.25}$ |
| VMR log(CO) | $-2.70^{+0.28}_{-0.48}$ | 4 | $-3.85^{+0.62}_{-0.87}$ |
| VMR log($CO_2$) | $-3.33^{+0.29}_{-0.25}$ | 43 | $-2.62^{+0.36}_{-0.34}$ |
| VMR log($CH_4$) | $-6.03^{+0.22}_{-0.20}$ | 4.2 | $-6.14^{+0.42}_{-1.70}$ |
| VMR log($SO_2$) | $-5.06^{+0.14}_{-0.15}$ | 5.5 | $-4.59^{+0.19}_{-0.18}$ |
| VMR log($H_2S$) | $-3.48^{+0.32}_{-1.35}$ | 2.0 | $-7.10^{+3.18}_{-3.00}$ |
| VMR log($NH_3$) | $-6.38^{+0.78}_{-1.54}$ | 1.5 | $-8.12^{+1.99}_{-2.36}$ |
| VMR log(HCN) | | | $-9.12^{+2.01}_{-1.94}$ |
| | ATMO Non-Equilibrium Forward | | |
| Metallicity, $Z/Z_\odot$ | $43\pm8$ | | |
| Intrinsic Temperature, $T_{int}$ (K) | $458\pm38$ | | |
| Diffusion coeff., log[$K_{zz}$ ($cm^2/s$)] | $11.6\pm0.1$ | | |

The retrieval results from the ATMO and NEMESIS codes are given, with the best-fit parameter values shown along with their 1-σ uncertainties. $T_{int}$ refers to the intrinsic temperature, which is related to the flux, $F$, emitted from the planet's interior by $F = \sigma_R T_{int}^4$, where $\sigma_R$ the Stefan-Boltzmann constant. VMR refers to the Volume Mixing Ratio.

**Extended Data Table 3 | Aerosol Cloud Retrieval Comparisons**

| aerosol model | parameters | model selection | VMR $CH_4$ | $\log[K_{zz}\ (cm^2/s)]$ |
|---|---|---|---|---|
| | ATMO Retrieval | $\Delta$BIC | | |
| grey cloud | $\delta_{cloud}$ | - | $-6.03^{+0.22}_{-0.20}$ | $11.60\pm0.11$ |
| cloud + haze | $\delta_{cloud}, \delta_{haze}, \alpha_{haze}$ | 5.0 | $-6.29^{+0.19}_{-0.26}$ | $11.56\pm0.09$ |
| grey cloud | $\log$(cloud top) | 5.3 | $-5.96^{+0.24}_{-0.25}$ | $11.47\pm0.11$ |
| | NEMESIS Retrieval | $\Delta(\ln BE)$ | | |
| grey cloud | $\log$(cloud top) | - | $-6.14^{+0.42}_{-1.70}$ | |
| grey cloud | $\delta_{cloud}$ | -4.1 | $-5.86^{+0.35}_{-0.85}$ | |
| enstatite cloud | $\delta_{cloud}$ | -20.0 | $-5.38^{+0.23}_{-0.27}$ | |

BIC refers to the Bayesian information criterion, while BE refers to the Bayesian Evidence. Both statistics are calculated relative to the best-fit model given a value of '-'.