## [Peer Review File · Nature]

Manuscript Title: A warm Neptune's methane reveals core mass and vigorous atmospheric mixing

Reviewer Comments & Author Rebuttals

Reviewer Reports on the Initial Version:

Referees' comments:

Referee #1 (Remarks to the Author):

This is an interesting and potentially very important paper, and appropriate for Nature. The data analysis, retrievals, and interior modeling are well done and seem sound to me. I have "one-and-a-half" major critiques, and many minor comments:

My principal critique is that the analysis merely assumes that the calculated equilibrium temperature of the planet is correct, and neglects a very simple and unconstrained possibility that the planet's atmospheric (not interior) temperature is higher than assumed. The abundance of methane in thermochemical equilibrium is highly sensitive to atmospheric temperature, and the calculated T_{eq} depends on the assumed degree of heat redistribution and the stellar irradiance. Even if the assumption of efficient heat redistribution is correct, there are uncertainties in the stellar temperature and radius, and the planet could arguably be hotter (or cooler) than calculated.

Under the assumption of efficient heat redistribution (that you implicitly adopt by using the published T_{eq}), the terminator region probed by transit spectroscopy should have approximately the same temperature as the sub-stellar point. Secondary eclipses probe the sub-stellar point and would be a good check on the atmospheric temperature profiles that you're using (Figure 9). JWST secondary eclipses would be best, but lacking those Spitzer could be helpful. Spitzer observed four secondary eclipses of this planet. The eclipse depths and brightness temperatures are in the large compilation from Deming+2023 (AJ, 165, 104, their Tables 1 and 2, via <https://vizier.cds.unistra.fr/viz-bin/VizieR>). I suggest calculating the depths of formation for the two Spitzer bands and adding those T_b values with error bars to Figure 9. Presuming that they agree with your $T(P)$ profile, they would "anchor" the profile in the atmosphere and add more credibility to your analysis.

My "half-major" critique is that you're using "temperature" very loosely, and many readers will be confused. Your term "intrinsic temperature" evidently is equivalent to an effective temperature in the radiative transfer sense, i.e. it's the temperature of an equivalent blackbody that drives the required flux from the interior. But you don't say that, and it needs to be made very clear and explicit (many Nature readers are not astronomers). And the Tables and Figures use "temperature" inconsistently. Table 2 and Figure 7 seem to list intrinsic temperature (but as simply "temperature"), whereas Figure 8 is illustrating a retrieval of atmospheric temperature at a specific depth. All of those differences need to be cleaned up and made consistent, and terms clearly defined.

Minor comments:

Line 57: Our abundance constraints are highly constraining >> Our results are highly constraining (because it sounds redundant to say that constraints are constraining)

line 62, "but is able account" >> but is able to account

line 100: at a typical resolution >> at a typical spectral resolving power (resolution and resolving power are inversely related)

line 135: I presume this is the Q-branch that you're seeing? It would be informative to mention that the Q-branch of this band has the greatest intrinsic (quantum-mechanical) strength of any feature for methane (I think). So if CH₄ is detectable anywhere, it would be here.

Figure 1: The star spot crossing is not obvious on this Figure, and readers who look for it may be initially confused. I think you could omit mention of the spot crossing until later, such as when discussing Figure 6. Or refer to Figure 6 in the caption of Figure 1.

line 229 "silicates cloud particles" >> silicate cloud particles

line 231 refers to T_{eq} as 750K, whereas line 73 says 770K, a minor inconsistency.

line 232, I think that statement about silicate clouds being at unobservable pressures needs a citation (although I do believe it).

line 241: "core significant detected " >> do you mean a core significantly detected, or a significant core detected? Please fix the expression to clarify.

line 244 "Assuming a uniform composition core" > what does that mean? It needs more explanation, at least briefly here.

line 249 "which do no predict" >> which do not predict

Table 1, under Period the Table says "Reference needed". Please add that reference. And under Description, insert "Orbital period in days".

line 277, "with resolutions near 3000" >> That's resolving power, not resolution. And I suggest quoting the resolving power at both edges of the wavelength range - given on STScI's web site.

line 288, "cleaning of, bad pixels," >> the first comma is incorrect. So: cleaning of bad pixels, etc.

line 289, "we measured the positional shift of the spectral trace across the detector and shift-stabilized the images with flux-conserving interpolation." >> I presume you're discussing shifts in the wavelength dimension. I don't understand why interpolation is needed, and I think it might even be detrimental: it's dangerous to interpolate data when the extracted signal is small, and arguably better to interpolate the extraction window rather than the data. If the detector flat field has been applied (and is accurate), the flux in a spectral element can be extracted by moving the extraction window to compensate for the positional shifts. (There are probably small inaccuracies in the flat-field correction, but interpolation per se can not cure them.) Probably the interpolation is merely a convenience, to allow you to associate wavelengths with consistent pixel-positions (?). Some clarification is needed as per exactly why the interpolation is necessary.

Figure 5: fix the "andq2" in the second line of the caption"

line 391: Nested >> nested

line 395: compared the >> compared to the

line 407: an simple isotherm >> a simple isotherm

Caption of Figure 7. You only need to say "posterior" once. But it's posteriors (plural). And the current caption is not very informative, isn't there more that should be said here? (Like, what are the contours? And some readers may need to be reminded that VMR is volume mixing ratio.)

Referee #2 (Remarks to the Author):

Report on "JWST NIRSpec reveals vigorous atmospheric mixing and the interior of a warm Neptune" by Sing et al., submitted to Nature.

The manuscript analyzes JWST NIRSpec observations of exoplanet WASP-107b in transmission (during transit), finding evidence for an excess in SO₂ and a lack of CH₄ compared to equilibrium models. These departures are attributed to a very vigorous vertical mixing, several orders of magnitude more efficient than usually assumed. The detection of other species such as H₂O, CO₂ and CO leads to constraints on the atmospheric metallicity (about 43 times solar) and intrinsic temperature (around 460 K, more than 4 times that of Jupiter today). Combining the constraints obtained on the atmospheric properties to the interior models, the authors also derive a core mass of about ~10 Earth masses, more in line with expectations than another study of the same object that derived a 4.6 Earth masses 3 sigma upper limit.

The data at hand, the NIRSpec spectrum between 2.7 and 5.2 microns, showing evidence for the presence of CO₂ and SO₂, is impressive. The manuscript is clear and well written and the results, linking atmospheric composition to interior properties are far-reaching. On this basis, I find the manuscript definitely suitable for publication in Nature. However, at this point, I need to be convinced that the

results are robust, as I explain below.

My most major concern is linked to Fig. 2. The ability of JWST to provide a spectrum on such a long wavelength range is a game changer of course. One feature is striking here: The horizontal line used for the contribution of aerosols opacities. The fact that, in lack of better information, the authors go for a simple solution is natural. However, the question of the consequences of this simple solution is one that should be addressed. The problem actually arises for giant planets closer to us: On Jupiter, condensed particles are responsible for a mysterious opacity source at 3 microns, and generally the absorption slope depends quite significantly on the size and composition of the condensed particles -see Sromovsky & Fry (2010) or Grassi et al. (2021). Also, for WASP-107b itself, Dyrek et al. (2024) claim the detection of silicate clouds using MIRI - their fig. 2 showing the planet's transmission spectrum infers a variability of the cloud opacities that is of order 5% in R_p/R_* . For comparison, the manuscript's Fig. 2 has a range of R_p/R_* that spans only 9%. Even if we leave aside the possibility of large absorption features of condensed particles like the 3 micron feature in Jupiter, for WASP-107b, the question remains whether a slope in the aerosol opacity can be accommodated by the data at hand and what are the consequences for the retrieved abundances.

A related question also concerns the zero level for the NRS1 and NRS2 parts of the spectra. What is the error on that one? It is probably small, but this should be quantified, probably using the residuals of the time series shown in Fig. 1.

Another major result of the work is the ability to constrain T_{int} , the planet's intrinsic temperature (or more precisely its intrinsic flux σT_{int}^4). Generally, because the planet flux σT_{int}^4 is significantly smaller than the irradiation flux σT_{eq}^4 , no useful constraint can be derived from the spectrum. This is not the case here because of the presence of disequilibrium species, particularly CO₂ and CH₄. As explained in the paper, the low but not negligible CH₄ abundance essentially puts a lower limit to the atmospheric temperature where CH₄ is quenched. An upper limit is derived from the abundance of other species, i.e., CO₂ (and presumably CO). However, this requires a series of assumptions that should be discussed: (1) The atmosphere can be considered 1D, i.e. independent of latitude and longitude. (2) T_{int} and (3) K_{zz} are assumed to be uniform with depth.

Given that WASP-107b is cooler than a typical hot Jupiter, 3D effects are probably not dominant, so that (1) shouldn't be problematic. Hypothesis (2) requires that whatever mechanism is responsible for the heat dissipation, it takes place in the deep interior. If this hypothesis is lifted however, then a new class of atmospheric solutions have to be derived - including the possibility of low-pressure convective zones (see e.g. Guillot & Showman 2002), affecting the very definition of T_{int} and the link between atmospheric and interior constraints. Similarly, if separate convective zones appear, hypothesis (3) should be revised. These possibilities should be discussed, and the necessary caveats provided.

A last minor comment is that in Fig. 1 (bottom panel) the colors chosen for NRS1 and NRS2 are barely distinguishable.

Author Rebuttals to Initial Comments:

We thank the referees and the editor for their constructive comments. Below we have detailed our responses to each of the major and minor points raised, with the comment in blue and response given in black. In response, we have updated the manuscript accordingly. The changes are noted in the responses, and we have also bolded altered text in the manuscript to more easily see the associated changes associated with the referee responses.

In response to the editor, the main section of the paper has been substantially shortened to conform to Nature's requirements. This required considerable re-arranging of the main text, with some text also moved to the methods section. Other requested editorial changes have also been made, including a shortened title.

Referees' comments:

Referee #1 (Remarks to the Author):

This is an interesting and potentially very important paper, and appropriate for Nature. The data analysis, retrievals, and interior modeling are well done and seem sound to me. I have "one-and-a-half" major critiques, and many minor comments:

My principal critique is that the analysis merely assumes that the calculated equilibrium temperature of the planet is correct, and neglects a very simple and unconstrained possibility that the planet's atmospheric (not interior) temperature is higher than assumed. The abundance of methane in thermo-chemical equilibrium is highly sensitive to atmospheric temperature, and the calculated T_{eq} depends on the assumed degree of heat redistribution and the stellar irradiance. Even if the assumption of efficient heat redistribution is correct, there are uncertainties in the stellar temperature and radius, and the planet could arguably be hotter (or cooler) than calculated.

Under the assumption of efficient heat redistribution (that you implicitly adopt by using the published T_{eq}), the terminator region probed by transit spectroscopy should have approximately the same temperature as the sub-stellar point. Secondary eclipses probe the sub-stellar point and would be a good check on the atmospheric temperature profiles that you're using (Figure 9). JWST secondary eclipses would be best, but lacking those Spitzer could be helpful. Spitzer observed four secondary eclipses of this planet. The eclipse depths and brightness temperatures are in the large compilation from Deming+2023 (AJ, 165, 104, their Tables 1 and 2, via <https://vizier.cds.unistra.fr/viz-bin/VizieR>). I suggest calculating the depths of formation for the two Spitzer bands and adding those T_b values with error bars to Figure 9. Presuming that they agree with your $T(P)$ profile, they would "anchor" the profile in the atmosphere, and add more credibility to your analysis.

We agree with the referee that ruling out an anomalously high temperature of the planet is important, with the Spitzer observations placing a good empirical constraint on the atmospheric temperatures. We thank the referee for pointing out the Deming et al. reference. We have added their brightness temperatures onto Fig. 9, with the 4.5 micron brightness temperature of 947^{+91}_{-70} K and 670^{+147}_{-84} K for 3.6 microns found. The pressures were calculated from secondary eclipse model contribution functions integrated over the Spitzer bandpasses, with the model calculated from our best-fitting abundances and the $T_{int}=450$ K T-P profile. As can be seen in Fig. 9 (shown below for convenience), the Spitzer data clearly rules out hot atmospheric temperatures (≥ 1038 K at 1-sigma) and match well with our model T-P profile.

For reference, we also estimated the maximum equilibrium temperatures taking into account uncertainties in the orbit and stellar effective temperature (T_{eq} does not depend on the stellar radius). In the limit of a uniform planetary temperature and zero-albedo, we calculate $T_{eq} = 735$ K using the stellar $T_{eff}=4425\pm 70$ K (Piaulet et al. 2021) and the a/R_s we measure from the JWST data (Table 1), with a 2-sigma upper limit of $T_{eq}=760$ K (errors dominated by the stellar effective temperature uncertainty). The theoretical limit of the dayside temperature in the no-albedo, no-circulation limit is 940 ± 15 K, with the limb necessarily being lower than this temperature. These theoretical temperatures are in the range of the Spitzer 670 to 947 K brightness temperatures.

Both the theoretical maximum value and Spitzer data show the atmospheric temperature of the planet is confidently less than about 950 K. Thus, we conclude the hot atmospheric temperatures needed to explain a low CH_4 abundance from thermo-chemical equilibrium can be ruled out both theoretically and observationally, from independent data sources. Fig. 9 has been updated accordingly along with the figure caption. We feel the good

observational agreement indeed helps add to credibility of the model T-P profile, and we thank the referee for the comment and suggestion.

Figure 9 updated with the Spitzer brightness temperatures.

My "half-major" critique is that you're using "temperature" very loosely, and many readers will be confused. Your term "intrinsic temperature" evidently is equivalent to an effective temperature in the radiative transfer sense, i.e. it's the temperature of an equivalent blackbody that drives the required flux from the interior. But you don't say that, and it needs to be made very clear and explicit (many Nature readers are not astronomers). And the Tables and Figures use "temperature" inconsistently. Table 2 and Figure 7 seem to list intrinsic temperature (but as simply "temperature"), whereas Figure 8 is illustrating a retrieval of atmospheric temperature at a specific depth. All of those differences need to be cleaned up and made consistent, and terms clearly defined.

We thank the referees for pointing out the inconsistencies and vagueness when referring to the atmospheric temperature and intrinsic temperature. We added a formal definition of T_{int} into the manuscript and have clarified the text and table to be more explicit which temperature we are referring to.

Minor comments:

Line 57: Our abundance constraints are highly constraining >> Our results are highly constraining (because it sounds redundant to say that constraints are constraining)

Changed

line 62, "but is able account" >> but is able to account

Changed

line 100: at a typical resolution >> at a typical spectral resolving power (resolution and resolving power are inversely related)

Changed

line 135: I presume this is the Q-branch that you're seeing? It would be informative to mention that the Q-branch of this band has the greatest intrinsic (quantum-mechanical) strength of any feature for methane (I think). So if CH4 is detectable anywhere, it would be here.

Yes, the methane feature is the strong Q-branch 3.3 micron methane feature. We note this fact at the referenced line in the manuscript.

Figure 1: The star spot crossing is not obvious on this Figure, and readers who look for it may be initially confused. I think you could omit mention of the spot crossing until later, such as when discussing Figure 6. Or refer to Figure 6 in the caption of Figure 1.

Changed, we now reference Fig. 6 in the caption as suggested.

line 229 "silicates cloud particles" >> silicate cloud particles

Changed

line 231 refers to T_{eq} as 750K, whereas line 73 says 770K, a minor inconsistency.

Temperature changed to 770 K.

line 232, I think that statement about silicate clouds being at unobservable pressures needs a citation (although I do believe it).

Reference added. We additionally removed the “and metallicity near solar” statement, as it is not needed. Fortney (2020, AJ, 160, 288, Fig. 22) shows condensation curves for a WASP-107b model at 100x solar, showing the clouds forming at high pressures at low Tint.

line 241: "core significant detected " >> do you mean a core significantly detected, or a significant core detected? Please fix the expression to clarify.

The expression has been changed to “core significantly detected”

line 244 "Assuming a uniform composition core" > what does that mean? It needs more explanation, at least briefly here.

We have clarified the uniform composition in the text adding, “an isothermal 50-50% mixture of rock and water”

line 249 "which do no predict" >> which do not predict

Changed

Table 1, under Period the Table says "Reference needed". Please add that reference. And under Description, insert "Orbital period in days".

Reference added

line 277, "with resolutions near 3000" >> That's resolving power, not resolution. And I suggest quoting the resolving power at both edges of the wavelength range - given on STScI's web site.

Changed

line 288, "cleaning of, bad pixels," >> the first comma is incorrect. So: cleaning of bad pixels, etc.

Changed

line 289, "we measured the positional shift of the spectral trace across the detector and shift-stabilized the images with flux-conserving interpolation." >> I presume you're discussing shifts in the wavelength dimension. I don't understand why interpolation is needed, and I think it might even be detrimental: it's dangerous to interpolate data when the extracted signal is small, and arguably better to interpolate the extraction window rather than the data. If the detector flat-field has been applied (and is accurate), the flux in a spectral element can be extracted by moving the extraction window to compensate for the positional shifts. (There are probably small inaccuracies in the flat-field correction, but interpolation per se can not cure them.) Probably the interpolation is merely a convenience, to allow you to associate wavelengths with consistent pixel-positions (?). Some clarification is needed as per exactly why the interpolation is necessary.

We agree with the referee, that in general one could run into problems with the interpolation if the shifts were large. However, with CV3 data Rustamkulov et al. (2022, ApJ, 928, 7) found that pre-aligning the data led to fewer position-dependent correlations in the lightcurves compared to not-shifting the data. In any case, the shifts are extremely small as the on-sky pointing of JWST is excellent with the data aligned to about 1/1000 of a pixel. These small shifts mean the interpolations are highly linear, and the magnitude is small. We verified this statement with the NRS2 data, where we find the pixel-level light curves and precisions agree to within 1 ppm if we move the extraction window or pre-align the data.

We have clarified the text on this point in the methods.

Figure 5: fix the "andq2" in the second line of the caption"

Fixed

line 391: Nested >> nested

Changed

line 395: compared the >> compared to the

Changed

line 407: an simple isotherm >> a simple isotherm

Changed

Caption of Figure 7. You only need to say "posterior" once. But it's posteriors (plural). And the current caption is not very informative, isn't there more that should be said here? (Like, what are the contours? And some readers may need to be reminded that VMR is volume mixing ratio.)

The caption has been updated to indicate what the contours mean and define the volume mixing ratio.

Referee #2 (Remarks to the Author):

Report on "JWST NIRSpec reveals vigorous atmospheric mixing and the interior of a warm Neptune" by Sing et al., submitted to Nature.

The manuscript analyzes JWST NIRSpec observations of exoplanet WASP-107b in transmission (during transit), finding evidence for an excess in SO₂ and a lack of CH₄ compared to equilibrium models. These departures are attributed to a very vigorous vertical mixing, several orders of magnitude more efficient than usually assumed. The detection of other species such as H₂O, CO₂ and CO leads to constraints on the atmospheric metallicity (about 43 times solar) and intrinsic temperature (around 460 K, more than 4 times that of Jupiter today). Combining the constraints obtained on the atmospheric properties to the interior models, the authors also derive a core mass of about ~10 Earth masses, more in line with expectations than another study of the same object that derived a 4.6 Earth masses 3 sigma upper limit.

The data at hand, the NIRSpec spectrum between 2.7 and 5.2 microns, showing evidence for the presence of CO₂ and SO₂, is impressive. The manuscript is clear and well written and the results, linking atmospheric composition to interior properties are far-reaching. On this basis, I find the manuscript definitely suitable for publication in Nature. However, at this point, I need to be convinced that the results are robust, as I explain below.

My most major concern is linked to Fig. 2. The ability of JWST to provide a spectrum on such a long wavelength range is a game changer of course. One feature is striking here: The horizontal line used for the contribution of aerosols opacities. The fact that, in lack of better information, the authors go for a simple solution is natural. However, the question of the consequences of this simple solution is one that should be addressed. The problem actually arises for giant planets closer to us: On Jupiter, condensed particles are responsible for a mysterious opacity source at 3 microns, and generally the absorption slope depends quite significantly on the size and composition of the condensed particles -see Sromovsky & Fry (2010) or Grassi et al. (2021). Also, for WASP-107b itself, Dyrek et al. (2024) claim the detection of silicate clouds using MIRI - their fig. 2 showing the planet's transmission spectrum infers a variability of the cloud opacities that is of order 5% in R_p/R^ . For comparison, the manuscript's Fig. 2 has a range of R_p/R^* that spans only 9%. Even if we leave aside the possibility of large absorption features of condensed particles like the 3 micron feature in Jupiter, for WASP-107b, the question remains whether a slope in the aerosol opacity can be accommodated by the data at hand and what are the consequences for the retrieved abundances.*

We have added further details to the retrievals to better illustrate how accommodating the data can be to different aerosol assumptions and how it affects the abundances. Table 3 has been added to help summarise these results (shown below for convenience), with the CH₄ abundance listed given it's the most relevant for interpreting the non-eq chemistry. When adding a parameterized aerosol slope (Eq 3) along with a grey cloud, we find the slope is not statistically justified by the Bayesian Information Criteria. In addition, the resulting abundance for CH₄ is also not significantly affected. Thus, we have no evidence for a significant slope in the NIRSpec data, and the abundances in any case are not affected by the inclusion of a slope. With NEMESIS, we also performed a retrieval where enstatite clouds were assumed, based on the MIRI Dyrek et al. (2024) findings. The enstatite cloud was also not statistically favoured, with the CH₄ abundance consistent at the 2-sigma level with the nominal model used in the manuscript.

aerosol model	parameters	model selection	VMR CH ₄
	ATMO Retrieval	Δ BIC	
grey cloud	δ_{cloud}	-	-6.03 ^{+0.22} _{-0.20}
cloud + haze	$\delta_{\text{cloud}}, \delta_{\text{haze}}, \alpha_{\text{haze}}$	5.0	-6.29 ^{+0.19} _{-0.26}
grey cloud	log(cloud top)	5.3	-5.96 ^{+0.24} _{-0.25}
	NEMESIS Retrieval	Δ (lnBE)	
grey cloud	log(cloud top)	-	-6.14 ^{+0.42} _{-1.70}
grey cloud	δ_{cloud}	-4.1	-5.86 ^{+0.35} _{-0.85}
enstatite cloud	δ_{cloud}	-20.0	-5.38 ^{+0.23} _{-0.27}

Extended Data Table 3 Aerosol Cloud Retrieval Comparisons. BIC refers to the Bayesian information criterion, while BE refers to the Bayesian Evidence. Both statistics are calculated relative to the best-fit model given a value of '-'.

A related question also concerns the zero level for the NRS1 and NRS2 parts of the spectra. What is the error on that one? It is probably small, but this should be quantified, probably using the residuals of the time series shown in Fig. 1.

The baseline flux in the NRS1 and NRS2 white light curves (which we assume the referee is referring to) is measured to 4.5 and 12.5 ppm respectively. This precision is well below the per-point error levels of the transmission spectra, which typically have ~100 ppm error bars per point in the 3.5 to 4 micron region. There is also no evidence for a significant offset between the NRS1 and NRS2 detectors to near the white light curve precision levels. If one measures the difference from the transmission spectra averaging together the red side of NRS1 (3.6 to 3.7 microns) and comparing it to the blue side of NRS2 (3.86 to 3.96 microns), we find a 15+/- 33 ppm difference.

To further corroborate the data reduction, we also added an independent reduction of the data detailed in the methods subsection TEATRO. The spectra agree between TEATRO and the FIREFLY reduction at better than 1-sigma, indicating the spectral shape and NRS1/NRS2 zero point levels are largely independent of data reduction methods.

Another major result of the work is the ability to constrain T_{int} , the planet's intrinsic temperature (or more precisely its intrinsic flux σT_{int}^4). Generally, because the planet flux σT_{int}^4 is significantly smaller than the irradiation flux σT_{eq}^4 , no useful constraint can be derived from the spectrum. This is not the case here because of the presence of disequilibrium species, particularly CO₂ and CH₄. As explained in the paper, the low but not negligible CH₄ abundance essentially puts a lower limit to the atmospheric temperature where CH₄ is quenched. An upper limit is derived from the abundance of other species, i.e., CO₂ (and presumably CO). However, this requires a series of assumptions that should be discussed: (1) The atmosphere can be considered 1D, i.e. independent of latitude and longitude. (2) T_{int} and (3) K_{zz} are assumed to be uniform with depth.

Given that WASP-107b is cooler than a typical hot Jupiter, 3D effects are probably not dominant, so that (1) shouldn't be problematic. Hypothesis (2) requires that whatever mechanism is responsible for the heat dissipation, it takes place in the deep interior. If this hypothesis is lifted however, then a new class of atmospheric solutions have to be derived - including the possibility of low-pressure convective zones (see e.g. Guillot & Showman 2002), affecting the very definition of T_{int} and the link between atmospheric and interior constraints. Similarly, if separate convective zones appear, hypothesis (3) should be revised. These possibilities should be discussed and the necessary caveats provided.

We agree with the referee that these assumptions should be more clearly stated, and caveats discussed. We have added the following paragraph to the methods in non-equilibrium model section to address this point.

Our use of a 1D atmospheric model to interpret the non-equilibrium chemistry has several assumptions. The first is that the atmosphere can be considered 1D and is independent of both latitude and longitude. Hot and ultra-hot Jupiters show large day-night temperature gradients [87]. Spitzer phase curves show these differences decrease at lower T_{eq} [88]. Thus, the latitude and longitude temperature differences for warm ~750K planets like WASP-107b are expected to be modest. Another important assumption is that the parameters of T_{int} and K_{zz} are constant and assumed to be uniform with depth. Our model contains a single convective region (see Fig. 9), though the possibility of a separate low-pressure convective zone has been studied [89]. Multiple convective regions would affect the interpretation of T_{int}

and the link between atmospheric and interior constraints. In addition, our modeling also assumes a single K_{zz} value which is constant with altitude. Theoretical studies have predicted that K_{zz} increases with altitude[14]. Higher K_{zz} values at higher altitudes would not greatly affect our results, as the quench pressures for the molecular features probed here are similar (see Fig. 9). However, if multiple convective zones are present, the use of a single K_{zz} value may have to be re-visited.

A last minor comment is that in Fig. 1 (bottom panel) the colors chosen for NRS1 and NRS2 are barely distinguishable.

The figure colors were hard to distinguish. We have altered the colors in Fig. 1 to better differentiate the data and models.

Reviewer Reports on the First Revision:

Referees' comments:

Referee #1 (Remarks to the Author):

The authors have done a good job responding to my comments, and I think this paper is (almost) ready for publication in Nature. I don't need to review it again, but I did notice two last minor issues: 1) the comment the authors added to the caption of Figure 1 refers to Figure 3, but I think they mean Extended Data Figure 3, and 2) line 350, spell across, not accross.

This is a very interesting paper, and I enjoyed reviewing it.

Referee #2 (Remarks to the Author):

I am globally satisfied with the changes to the manuscript and the authors' responses to the reviewers. However, one last point remains: The authors have presented in Table 3 different possibilities including solutions not limited to a grey cloud. They show that these solutions lead to different values of the CH₄ abundance, generally within 2 sigma from their preferred model. Given that the authors derive values of K_{zz} 3 to 4 orders of magnitude larger than those usually derived from GCM simulations, they should also provide the values of K_{zz} that are inferred from solutions with these other cloud models.

Author Rebuttals to First Revision:

We thank the referees and editor for their efforts in reviewing the paper and the constructive comments. The manuscript has been revised, taking into account the final comments each referee has. More detailed responses to each point are found below. We have also further revised the manuscript to conform to the Nature editorial process. The first paragraph after the bold text has been condensed into a single added sentence in the bold text. All the figures and tables have also been removed, leaving only the captions for both.

We would like to go down the AAP path. If I understand the process correctly, the manuscript would come out online about 6 days after acceptance (so perhaps April 8th or so?), which would put a target publication date three weeks after submission, so about April 25th. If I'm understanding the process, then we can proceed with these dates.

I wish to participate in transparent peer review.

Referee Responses:

Referee #1. The authors have done a good job responding to my comments, and I think this paper is (almost) ready for publication in Nature. I don't need to review it again, but I did notice two last minor issues: 1) the comment the authors added to the caption of Figure 1 refers to Figure 3, but I think they mean Extended Data Figure 3, and 2) line 350, spell across, not accross.

We thanks the reviewer for pointing out these mistakes. Both issues have been addressed in the revised manuscript.

I am globally satisfied with the changes to the manuscript and the authors' responses to the reviewers. However, one last point remains: The authors have presented in Table 3 different possibilities including solutions not limited to a grey cloud. They show that these solutions lead to different values of the CH₄ abundance, generally within 2 sigma from their preferred model. Given that the authors derive values of K_{zz} 3 to 4 orders of magnitude larger than those usually derived from GCM simulations, they should also provide the values of K_{zz} that are inferred from solutions with these other cloud models.

Referee #2. I am globally satisfied with the changes to the manuscript and the authors' responses to the reviewers. However, one last point remains: The authors have presented in Table 3 different possibilities including solutions not limited to a grey cloud. They show that these solutions lead to different values of the CH₄ abundance, generally within 2 sigma from their preferred model. Given that the authors derive values of K_{zz} 3 to 4 orders of magnitude larger than those usually derived from GCM simulations, they should also provide the values of K_{zz} that are inferred from solutions with these other cloud models.

We agree these K_{zz} values are a good addition to the table. We have added these values to Extended Data Table 3 to the ATMO retrievals. In each case, the inferred K_{zz} values are within 2-sigma of the adopted K_{zz} value reported in the text.